# Modeling the Changes in Water Balance Components of Highly Irrigated Western Part of Bangladesh

A.T.M. Sakiur Rahman[1*], Md. Shakil Ahmed[2], Hasnat Mohammad Adnan[3], Mohammad Kamruzzaman[4], Md. Abdul Khalek[2], Quamrul Hasan Mazumder[3] and Chowdhury Sarwar Jahan[3]

**Authors details:**

[1]Hydrology Lab, Graduate School of Science and Technology, Kumamoto University, 2-40-1 Kurokami, Kumamoto, Japan
[2]Department of Statistics, University of Rajshahi, Rajshahi 6205, Bangladesh
[3]Department of Geology and Mining, University of Rajshahi, Rajshahi 6205, Bangladesh
[4]Institute of Bangladesh Studies, University of Rajshahi, Rajshahi 6205, Bangladesh

*Corresponding author e-mail: shakigeo@gmail.com

**Abstract.** The objectives of the present study are to explore the changes in water balance components (WBCs) by co-utilizing Discrete Wavelet Transformation (DWT) and different forms of Mann–Kendal (MK) test and to develop Wavelet Denoise Autoregressive Integrated Moving Average (WD-ARIMA) models for forecasting the WBCs. The results reveal that most of the trends (about 73%) identified in potential evapotranspiration ($P_{ET}$) have decreasing tendency during the hydrological years 1981-82 to 2012-13 in the western part of Bangladesh. However, most of the trends (about 82%) are not statistically significant at 5% level of significance. Actual evapotranspiration ($A_{ET}$), annual deficit and annual surplus also show almost similar tendency. Rainfall and temperature show increasing trends, but WBCs show inverse tendency suggesting traditional concept of change in $P_{ET}$ associated with changes in temperature, those cannot explain the change in WBCs. Moreover, it is found that generally 8-year (D3) to 16-year (D4) periodic components are effective components and are responsible for trends found in original data of WBCs in the area. The wavelet denoising of WBCs time series has been done to improve the performance of ARIMA model as actual data affected by noise and show unsatisfactory model performance. The quality of denoising time series data has been ensured by relevant statistical analysis. The performance of WD-ARIMA model has been assessed by Nash–Sutcliffe Efficiency ($NSE$) coefficient and coefficient of determination ($R^2$). The WD-ARIMA model shows acceptability with very good performance that clearly demonstrates the advantages of denoising of the time series data for forecasting WBCs. The validation results of models reveal that the forecasted values are very close to actual ones with acceptable mean percentage error, and residuals also follow the normal distribution. The performance and validation results indicate that models can be used for short-term forecasting of WBCs. Further studies on different combinations of wavelet analysis would be facilitated to develop better model for hydrological forecasting in context of climate change, and findings of the study can be used to improve the water resources management in highly irrigated western part of Bangladesh.

**Keywords:** Discrete Wavelet Transformation, Wavelet Denoising, Water Balance, ARIMA Model

## 1. Introduction

After introducing the monthly water balance model by Thronthwaite (1948) and afterward followed by Thornthwaite and Mather (1957), this model is going through modifications for adaptation in the different areas of the world. The development of the new model is still ongoing (Xu and Singh, 1998) as the water balance model is significantly important in water resources management, irrigation scheduling and crop pattern designing (Kang et al., 2003; Valipour, 2012). Moreover, it can be used for the reconstruction of catchment hydrology, climate change impact assessment and streamflow forecasting (e.g. Alley, 1985; Arnall, 1992, Xu and Halldin, 1996; Molden and Sakthivadivel, 1999; Boughton, 2004; Anderson et al., 2006; Healy et al., 2007; Moriarty et al., 2007; Karimi et al., 2013). Therefore, detecting the changes in WBCs and more accurate forecasting of WBCs are important for achieving the sustainability of water resources management. However, hydro-meteorological time series are contaminated by noises from hydro-physical processes that affect the accuracy of analysis, simulation and forecasting (Sang et al., 2013 and Wang et al., 2014). Hence, it is necessary to denoise the time series for improving the accuracy of the obtained results. In the present study, wavelet denoising technique has been coupled with ARIMA models for forecasting the WBCs after detecting the changes in WBCs by different forms of MK tests and identifying the time period responsible for trends in WBCs time series using DWT time series data.

Generally, physics based numerical models are used for understanding a particular hydrological system and forecasting the water balance or budget (e.g. Fulton et al., 2015, Leta et al., 2016) components. In this method, for reliable forecasting, a large amount of hydrological data is required to assign physical properties of the grid and model parameters and to calibrate the model simulation. However, they have a number of limitations in practice including the cost, time and availability of the data (Yoon et al., 2011; Adamowski and Chan, 2011). Data based forecasting models, statistical models, are suitable alternatives to overcome these problems. The most common statistical methods for hydrological forecasting are ARIMA models and multiple linear regression (Young, 1999; Adamowski, 2007). Many studies use ARIMA model to predict water balance input parameters like rainfall (e.g., Rahman et al., 2015; Rahman et al., 2016), temperature (e.g. Nury et al., 2016) and $P_{ET}$ (e.g., Valipour, 2012). However, ARIMA model cannot handle non-stationary hydrological data without pre-processing of the input time series data (Tiwari and Chatterjee, 2010; Adamowski and Chan, 2011). Wavelet analysis, a new method in the area of hydrological research, is such a method that is able to handle non-stationary data effectively (Adamowski and Chan, 2011). However, over the course of time some research works have already been done. For example, Adamowski and Chan (2011) coupled wavelet analysis with Artificial Neural Network (ANN) models for forecasting the hydrological variables like groundwater level in Quebec, Canada. Kisi (2008) and Partla (2009) and Santos and da Silva (2014) develop a hybrid wavelet ANN models for monthly and daily streamflow forecasting respectively. A study conducted by Rahman and Hasan (2014) also finds that the performance of the wavelet-based ARIMA models is better than the classical ARIMA model for forecasting the humidity of Rajshahi meteorological station in Bangladesh. A comparative study of wavelet ARIMA models and wavelet ANN models has been conducted by Nury et al. (2017). The study shows that the wavelet ARIMA models are more effective than the wavelet ANN for temperature forecasting. Khalek and Ali (2016) developed wavelet seasonal ARIMA (W-SARIMA) and neural network autoregressive (W-NNAR) model for forecasting the groundwater level. The study also finds that the performance of W-SARIMA model is better than the performance of W-NNAR models. All of these studies mentioned above find that the performance of wavelet aided model is better than classical ARIMA models and ANN models. Moreover, the

analysis of periodicity using wavelet transformed details, and approximation components of hydro-
meteorological time series data can better provide insight into trends and effects of time period on trend (e.g.
Nalley et al., 2013; Araghi et al., 2014; Pathak et al., 2016). As a result, wavelet transformation of hydro-
meteorological time series is gaining popularity in recent years to detect periodicity (e.g. Partal and Küçük,
2006; Partal, 2009; Nalley et al., 2013; Araghi et al., 2014; Pathak et al., 2016). Some studies have been
conducted on spatio-temporal characteristics of hydro-meteorological variables such as rainfall (e.g. Shahid and
Khairulmaini, 2009; McSweeney et al., 2010; Ahasan et al., 2010; Kamruzzaman et al., 2016a, Rahman and
Lateh, 2016; Rahman et al., 2016; Syed and Al Amin, 2016), temperature (e.g. Shahid, 2010; Nasher and Uddin,
2013; Rahman, 2016; Syed and Al Amin, 2016; Kamruzzaman et al., 2016a), $P_{ET}$ (Hasan et al., 2014; Acharjee,
2017) in Bangladesh. Karim et al. (2012) study the WBCs like $P_{ET}$, AET, deficit and surplus of water of 12
districts in Bangladesh and Kanoua and Merkel (2015) study the water balance of Titas Upazila (Sub-district) in
Bangladesh. So far, all studies carried out on hydrological variables in Bangladesh have the following
limitations: most of the studies were limited to detect trends or forecasting of rainfall and temperature and a few
studies on $P_{ET}$ and water balance. Therefore, the present study has been conducted to detect trends and to
identify periodicities in WBCs such as potential evapotranspiration ($P_{ET}$), actual evapotranspiration ($A_{ET}$),
annual deficit and surplus of water by co-utilizing DWT and different forms of Mann-Kendal (MK) test in the
western part of Bangladesh; and to develop WD-ARIMA models for forecasting the WBCs. To date, there is no
comprehensive study that couples wavelet denoising methods with ARIMA models for forecasting WBCs.
Wavelet denoising methods are widely used in many other engineering and scientific fields; however, they have
been little used in hydrology (Sang, 2013). Hence, it is expected that the new combinations will better explore
insight the water balance components which will ultimately help policymakers to prepare sustainable water
resources management plans.
**2. Study Area, Data and Methods**
**2.1 Study area**
Bangladesh enjoys a humid, warm and tropical climate. The western part of Bangladesh covers about 41% or
60,165 km$^2$ of the country. The geographic coordinates of the study area extends between 21°36′-26°38′N
latitude and 88°19′-91°01′E longitude. Annual rainfall and average temperature in the area vary from 1492 to
2766 mm with an average of 1925 mm and 24.18 to 26.17°C with an average of 25.44°C respectively
(Kamruzzaman et al., 2016a). Bangladesh is the fourth biggest rice producing country in the world (Scott and
Sharma, 2009) and the livelihoods of the majority of the people (about 75%, Shahid and Behrawan, 2008;
Kamruzzaman et al., 2016b) are related to agricultural practices. Crop calendar of Bangladesh is related to the
climatic seasons. Rice grows in three seasons (*Aus, Aman and Boro* seasons) in Bangladesh. Almost 73.94%
cultivable area is used for *Boro* rice cultivation in the country (Banglapedia, 2003). *Aus* and *Aman* rice are
mainly rain-fed crops; however, *Boro* rice is almost groundwater-fed (Ravenscroft et al., 2005) and requires
about 1m of water per square meter in Bangladesh (Harvey et al., 2006; Michael and Voss, 2009).
**2.2 Data**
National climate database of Bangladesh prepared by Bangladesh Agricultural Research Council (BARC) has
been used for the study. The database is available for research and can be found in BARC website
(http://climate.barcapps.gov.bd/). The database has been prepared from the data recorded by Bangladesh
Meteorological Division and contains long-term monthly climate data such as rainfall, minimum, maximum and
average temperatures, humidity, sunshine hours, wind speed and cloud cover. The locations of the
meteorological stations in the study area are shown in Figure 1. The data has been rearranged following the
hydrological year for the period 1981-82 to 2012-13. The hydrological year in Bangladesh starts in April and
ends in March.

## 2.3 Methods

In the present study, WBCs have been calculated and trends in WBCs have been identified by MK/MMK test
for evaluating the long-term water balance of the highly irrigated western part of Bangladesh. DWT data of
WBCs time series has been analyzed for identifying the time period responsible for the trend in the data. WBCs
have been forecasted by ARIMA models and the model performance has been evaluated statistically. If the
performance of the model is not satisfactory for forecasting the WBCs, the denoising of original time series has
been done using discrete wavelet transformation techniques to improve the performance of the model. The
descriptions of the methods have been presented in the following sections.

### 2.3.1 Calculation of Potential Evapotranspiration and Water Balance Components

Potential evapotranspiration ($P_{ET}$) is the key parameter to estimate WBCs. It has been calculated by Penman-
Monteith equation (Allen et al., 1998) in the present study. The soil-water balance concept proposed by
Thornthwaite and Mather (1955) is one of the most widely used methods for estimating the WBCs. It is suitable
for assessing the effectiveness of agricultural water resources management practices and regional water balance
studies as it allows estimating the actual evapotranspiration ($A_{ET}$), water deficit and surplus (e.g., Chapman and
Brown 1966, Bakundukize et al., 2011, Karim et al., 2012, Viaroli et al., 2017). $A_{ET}$ is the amount of water
which is removed from the surface due to the process of evaporation and transpiration. The amount by which
$P_{ET}$ exceeds $A_{ET}$ is termed as deficit and surplus is the excess rainfall after the soil has reached its water holding
capacity (de Jong and Bootsma, 1997). It is necessary to calculate the field capacity of the soil for estimating the
WBCs. Field capacity of soil in the study area has been calculated using the soil texture map of Bangladesh
prepared by Soil Resource Development Institute Bangladesh (SRDI, 1998) where the description of soils has
been presented by Huq and Shoaib (2013). The values for water holding capacity of soil and rooting depth of the
plants suggested by Thornthwaite and Mather (1957) have been used for WBCs estimation in the present study.
The first step of the calculation is the subtraction of 5% rainfall from the monthly rainfall data as this amount of
water has been lost due to direct runoff (Wolock and McCabe, 1999; Karim et al., 2012; Kanoua and Merkel,
2015). The remaining amount of rainfall has been included in the calculation. The WBCs like $A_{ET}$, surplus and
deficit have been estimated based on the formulas presented in Table 1 and details of WBCs calculation can be
found in Electronically Supplementary Martial (EMS).

### 2.3.2 Trend Test

In the present study, the trends in WBCs have been detected by non-parametric Mann–Kendall (MK) (Mann,
1945; Kendal, 1975) test as it shows better performance to identify trends in hydrological variables like rainfall
(e.g. Shahid, 2010), temperature (e.g. Kamruzzaman et al., 2016a), $P_{ET}$ (e.g. Kumar et al., 2016), soil moisture
(e.g. Tabari and Talaee, 2013), runoff (e.g. Pathak et al., 2016), groundwater level (e.g. Rahman et al., 2016),
water quality (e.g. Lutz et al., 2016) in comparison to the parametric test (Nalley et al., 2012). MK test cannot
appropriately calculate the test statistic (Z) due to underestimating the variance (Hamed and Rao, 1998) if there
is a significant serial correlation at lag-1 in the time series data (Yue et al., 2002). The lag-1 auto-correlation has
been checked before analyzing the time series data if there is a significant lag-1 auto-correlation at 5% level, the
Modified MK test (Hamed and Rao, 1998) has been applied instead of MK test. The estimated Z statistic of
MK/MMK test has been evaluated for the direction of the trend such as positive Z statistic to indicate increasing
trend and vice versa. Moreover, it also indicates the level of significance of the obtained trend, for example, if
the calculated Z statistic is equal to or greater than the tabulated value of Z statistic +1.96 that indicates a
significant positive trend at 95% confidence level or if it is equal to or less than -1.96 that indicates a significant
decreasing trend. Moreover, the sequential values of $u(t)$ statistic of MK test derived from the progressive
analysis of MK test (Sneyers, 1990), u(t) is similar to the Z statistic (Partal and Küçük, 2006), have been used
for investigating the change point detection. The magnitude of the change has been calculated by Sen's slope
estimator (Sen, 1968). There are many good explanations (notably Nalley et al., 2012) of these methods
mentioned in this section and details regarding these, furthermore, can be referred to Mann (1945); Sen (1968);
Kendall (1971); Hamed and Rao (1998); Sneyers (1990); Yue et al. (2002).

### 2.3.3 Wavelet Transform and Periodicity

The wavelet analysis has been used to identify periodicity in hydro-climatic time series data (e.g., Smith et al.,
1998; Azad et al., 2015; Nalley et al., 2012; Araghi et al., 2014; Pathak et al., 2016) for different parts of the
world. Wavelet transform (WT), a multi-resolution analytical approach, can be applied to analyze time series
data as it offers flexible window functions that can be changed over time (Nievergelt, 2001; Percival and
Walden, 2000). It can be applied to detect the periodicity in hydro-climatic time series data (Smith et al., 1998;
Pišoft et al., 2004; Sang, 2012; Torrence and Compo, 1998; Araghi et al., 2014; Pathak et al., 2016) and
produces better performances in comparison to traditional approaches (Sang, 2013). There are two main kinds of
wavelet transform such as continuous wavelet transform (CWT) and discrete wavelet transform (DWT). The
application CWT is complex, as it produces a lot of coefficients (Torrence and Compo, 1998; Araghi et al.,
2014), whereas DWT is simple and useful for hydro-climatic analysis (Partal and Küçük, 2006; Nalley et al.,
2012). The wavelet coefficients following the DTW with dyadic format can be calculated as (Mallat, 1989):

$$\psi_{m,n}\left(\frac{t-\tau}{s}\right) = s_0^{-m/2}\,\psi\left(\frac{t - n\,\tau_o\,s_0^m}{s_0^m}\right) \dots\dots\dots\dots\dots\dots\dots. (1)$$

Where $\psi$ is the mother wavelet, the integers m and n are wavelet dilation and translation respectively. Specified
fixed dilation step ($s_o$) is greater than 1 and $\tau_o$ is location parameter. For the practical application, the values of
parameters $s_o$ and $\tau_o$ are considered as 2 and 1 respectively (Partal and Küçük, 2006; Pathak 2016). After
substituting these values in equation (1), the DWT for a time series $x_i$ becomes:

$$W_{m,n} = 2^{-m/2}\sum_{i=0}^{N-1} x_i\,\psi(2^{-m}\,i - n) \dots\dots\dots\dots\dots\dots\dots\dots\dots (2)$$

Where W indicates wavelet coefficient at scale $s = 2^m$ and location $\tau = 2^m n$.

In the DWT, details (D) and approximations (A) time series can emerge from the original time series after passing through low-pass and high-pass filters respectively. While approximations are the high scale and low-frequency components, details are the low scale and high-frequency components. Successive, iterations have been performed to decompose the time series into their several lower resolution components (Mallat, 1989; Misiti et al., 1997). In the present study, four levels (D1-D4) of decompositions have been performed following the dyadic scales and referred as D1, D2, D3 and D4 which are corresponds to 2, 4, 8 and 16year periodicity. Daubechies wavelet has been used in the present study as it performs better in hydro-meteorological studies (Nalley et al., 2012, 2013; Ramana et al., 2013; Araghi et al., 2014). To confirm about the periodicity present in the time series, correlation coefficient (*Co*) between *u*(*t*) of original data, *u*(*t*) of decomposition (D) time series data and different models (D1+A……..D4+D3+A) time series data have been calculated and the obtained results have been compared accordingly (Partal and Küçük, 2006; Partal, 2009).

### 2.3.4 ARIMA Models

To identify the complex pattern in data and to project the future scenario, ARIMA model (Box and Jenkins, 1976) has been used in hydrological science (e.g. Adamowski and Chan, 2011; Valipour et al., 2013; Nury et al., 2017; Khalek and Ali, 2016). The method includes three terms: (1) an autoregressive process (AR) represented by order-p, (2) nonseasonal differences for non-stationary data termed as order-d and (3) moving average process (MA) represented by order-q. ARIMA model of order (p, d, q) can be written as:

$$\emptyset_p(L)\,(1 - L)^d Y_t = \theta_0 + \theta_q(L)\,U_t \ldots \ldots \ldots \ldots \ldots \ldots \ldots \ldots \ldots \ldots (3)$$

Where, $\theta_0$ and $U_t$ are the intercept and white process with zero mean and constant variance respectively. $\emptyset_p(L)$ stands for AR term $(1 - \emptyset_1 L - \cdots - \emptyset_p L^p)$ and $\theta_q(L)$ represents MA term $(1 - \theta_1 L - \cdots - \theta_p L^p)$.

### 2.3.5 Wavelet Denoising

Wavelet de-noising based on thresholds introduced by Donoho et al. (1995) has been applied to the hydro-meteorological analysis (Wang et al., 2005 and 2014; Chou, 2011). In the present study, three-steps of analysis has been done for denoising the time series data as follows:

1. Decomposing the time series data *x*(*t*) into *M* resolution level for obtaining the detail coefficients ($W_{j,k}$) and approximation coefficients using DWT.

2. The detail coefficients obtained from DWT (1 to M levels) have been treated with threshold ($T_j$) selection. There are soft threshold and hard threshold to deal with detail coefficients and to get decomposed coefficient. In the present study, soft threshold has been selected as it's performs better than hard (Wang et al., 2014; Chou, 2011):

Soft threshold processing: $\quad W'_{j,k} = \begin{cases} sgn(W_{j,k})\left(|W_{j,k}| - T_j\right) & |W_{j,k}| > T_j \\ 0 & |W_{j,k}| < T_j \end{cases}$

3. Details coefficients from 1 to *M* level and approximate coefficients at level *M* have been reconstructed to get denoising time series data.

It is also necessary to select the threshold value for denoising the data. In the present study, Universal threshold
(UT) method (Donoho and Johnstone, 1994) has been used for estimating the threshold value as it shows good
performance in analyzing hydro-meteorological data (Wang et al., 2005; Chou, 2011).

## 2.3.6 Assessment of Model Performance

There are several indicators to assess the performance of the models. Nash–Sutcliffe Efficiency (*NSE*) (Nash
and Sutcliffe, 1970) coefficient, a normalized goodness-of-fit statistic, is the most powerful and popular method
for measuring the performance of hydrological models (McCuen et al., 2006; Moussa, 2010; Ritter and Muñoz-
Carpena, 2013). To evaluate and make a comparison between ARIMA and WD-ARIMA model, *NSE* has been
used in the study. *NSE* can be calculated as (Nash and Sutcliffe, 1970):

$$NSE = 1 - \frac{\sum_{i=1}^{N}(O_i - P_i)^2}{\sum_{i=1}^{N}(O_i - \bar{O})^2} = 1 - \left(\frac{RMSE}{SD}\right)^2 \ldots\ldots\ldots\ldots\ldots\ldots (4)$$

Where, $N$, $O_i$ and $P_i$ are the sample size, number of observation and model estimates respectively and
$\bar{O}$ and $SD$ are the mean and standard deviation of the observed values. The performance of a model can be
evaluated based on *NSE* value as: very good ($NSE \geq 0.90$); good ($NSE = 0.80\text{-}0.90$); acceptable ($NSE \geq 0.65$);
and unsatisfactory ($NSE < 0.65$) (Ritter and Muñoz-Carpena, 2013). $E_{RMS}$ is the root mean square error that can
be calculated as:

$$E_{RMS} = \sqrt{\frac{\sum_{i=1}^{N}(O_i - P_i)^2}{N}} \ldots\ldots\ldots\ldots\ldots\ldots\ldots\ldots\ldots\ldots\ldots\ldots (5)$$

The coefficient of determination ($R^2$) is another goodness of fit test to measure the performance of the models.
The perfect fit of the model draws a line between the actual values and fitted values, where $R^2$ value is 1. If $y_i$ is
the observation data, $\hat{y}_i$ is the model forecasted values of $y_i$ and $N$ is the number of data point used, $R^2$ can be
given as (Sreekanth et al., 2009):

$$R^2 = 1 - \frac{\sum_{i=1}^{N}(y_i - \hat{y}_i)^2}{\sum_{i=1}^{N}(y_i)^2 - \frac{(\sum_{i=1}^{N} y_i)^2}{N}} \ldots\ldots\ldots\ldots\ldots\ldots\ldots\ldots\ldots (6)$$

Moreover, mean percentage error ($E_{MP}$) and mean error ($E_M$) have also been calculated to evaluate the validation
of the model for forecasting. $E_{MP}$ reveals the percentage of bias (larger or smaller) of forecasted data over the
actual counterparts (Khalek and Ali, 2016). $E_{MP}$ and $E_M$ can be calculated as follows:

$$E_{MP} = \left(\frac{1}{n}\sum_{t=1}^{n}\frac{Y_t(actual) - Y_t(forecasted)}{Y_t(actual)}\right) \times 100\% \ldots\ldots\ldots (7)$$

$$E_M = \frac{1}{n}\sum_{t=1}^{n}[Y_t(actual) - Y_t(forecasted)]^2 \ldots\ldots\ldots\ldots\ldots\ldots\ldots (8)$$

## 3. Results of Analysis

## 3.1 Exploratory Statistics of Water Balance Components

Mean annual $P_{ET}$ during the period of 1981-82 to 2012-2013 in the study area varies from 1228 to 1460 mm (Figure 2) with an average of 1338 mm. The higher $P_{ET}$ values are found in the central part of the area where the annual rainfall is lower, but the temperature is higher (Kamruzzaman et al., 2016a). The standard deviations of $P_{ET}$ vary from 205 mm (in Jessore station) to 41 mm (in Bhola station). The $A_{ET}$ value (average = 925 mm) is almost 31% less than the $P_{ET}$ value as during the dry months (Dec-May), soil moisture condition reaches in a critical stage and $A_{ET}$ value is much lower than $P_{ET}$. The annual surplus of water varies from 515 to 1277 mm with an average of 838 mm. According to Wolock and McCabe (1999), 50% of surplus water can be considered as runoff for the major parts of the world. The higher surplus amount of water has been found in the northern part of the area and along the coastal area. The annual deficit of water that mainly occurs during the dry season (Dec to May) varies from 329 to 556 mm with an average of 416 mm (Figure 2). The highest annual deficit of water found in Rajshahi which is located in the central western part of the area where the depth of groundwater below the ground surface increases rapidly (Shamsudduha et al., 2009; Rahman et al., 2016).

## 3.2 Trend and Periodicity in Water Balance Components

### 3.2.1 Potential Evapotranspiration

The MK test or MMK test based on the lag-1 auto-correlation has been applied to detect the trend in $P_{ET}$. Table-2 shows the Z statistic of MK or MMK test of original time series data of $P_{ET}$ and Z statistic of the decomposition time series (D1-D4), approximation (A) and model (D1+A…..D3+D4+A) time series. The estimated Z statistic of original data ranges from -2.07 (Satkhira station) to 2.37 (Bhola station). These two stations out of total eleven show significant trends in $P_{ET}$. The plots of sequential $u(t)$ statistic of SMK test of these two stations are shown in Figure 3 where the dashed lines correspond to 5% significance level ($\pm 1.96$). The decreasing trend in $P_{ET}$ in Satkhira station started in the year 1985-86 and a significant decreasing trend started in 1993-94 hydrological year, and the trend become reverse after 2007-08. However, the significant increasing trend in $P_{ET}$ of Bhola station has been started very recently after some fluctuation.

Most of the trends (73%) in $P_{ET}$ in the study are negative and statistically insignificant at 95% confidence level or 5% significance level. Moreover, Z statistic of approximation (A) time series obtained by DWT indicates decreasing trends in $P_{ET}$ in all stations. The calculated Z statistic of approximation (A) time series is about -1.80 after rounding the figure for all stations as A time series data of all stations show a similar pattern (Electronic Supplementary Material (ESM) Fig. S1) over the time. The magnitude of change in $P_{ET}$ ranges from -10.89 mm/year in Satkhira station to 1.67 mm/year in Bhola station (Figure 4). The MK or MMK test has also been applied to the decomposition time series and model time series generates from the combination of approximation and decomposition time series data (Table 2 represents results of four stations based on alphabetic order and the full Table can be found in ESM Table S1). To find out the dominant periodicity affecting the trends in $P_{ET}$, two steps of analysis have been done. Firstly, the Z statistic which is the closest to the Z statistic of original time series data has been found out from the values of Z statistic of different models and decomposition (D) time series data. Secondly, the correlation coefficients (Co) of pairs of data (such as Co between $u(t)$ statistics of SMK of the original time series data and $u(t)$ statistics of SMK of D time series data) have been estimated and found out the highest Co from the estimated Co values for different pairs (Table 2).For example, the Z statistic of D4 time series data of Barisal station is 0.76 which is the nearest to Z statistic (0.72)

of the original time series data among the different models (Table 2). Moreover, Z statistic of model
(D3+D4+A) time series data is 0.56 which is the second nearest value to original time series with the highest
correlation coefficient ($Co$ = 0.85). Again D4 is present, hence D4 (16-year) is the dominant periodic
components on the trend in original data. However, D3 has also effect on the trend in the data. Therefore, D4
(16-year) is the basic periodic component, but 8-year (D3) periodicity has also effect on the trend. An additional
example, Z (2.47) statistic of the original time series of Bhola station is the closest to Z (2.36) statistic of the
model (D2+D4+A) time series data. However, the values of the Z statistic of D2, D4, D2+A and D4+A time
series are 0.61, 1.20, 0.48 and 0.90 respectively, which are not close to the Z statistic of the original time series
data. Hence, it is not clear from the Z statistic which periodic component (D2/D4) is the basic periodic
component for the significant trend in the original data. To get a clear idea about the dominant periodic
component, $Co$ coefficient values have been analyzed. It is seen that the $Co$ between $u(t)$ statistic of SMK of
original time series data and $u(t)$ statistic of SMK of D4 time series data is higher than the $Co$ between $u(t)$
statistic of SMK of original time series data and $u(t)$ statistic of SMK of D2 time series data (Table 2).
Moreover, Moreover, values of Z statistic of time series with D4 components like D4 and (D4+A) model time
series are higher than time series with D2 component (D2 and D2+A) (Table 2). It is, therefore, clear that D4 is
the main periodic component responsible for the trend in $P_{ET}$ data of Bhola station. However, Z statistic of D4 or
D4+A is not close to the Z statistic of original data (Table 2). Moreover, there is a statistically significant
positive trend in original data of $P_{ET}$ of Bhola station, but the trends of D4 and (D4+A) model time series data
are not statistically significant. When D2 time series add with (D4+A) model time series data, the Z statistic of
the resultant (D2+D4+A) model time series data becomes very close to original time series data. The trend of
(D2+D4+A) model time series is also statistically significant like the trend in original time series data (Table 2).
Hence, D2 has also effect on the trend in the original time series data. Station-wise analysis indicates that almost
half of the stations show the harmoniousness between the Z statistic of (D3+D4+A) model and original time
series data. When D3 and D4 time series have been analyzed separately, it is found that the higher relationship
exists between D4 and original time series data. Again, three stations (Dinajpur, Ishurdi and Jessore) show the
similarity in estimated Z statistic of original and (D1+D4+A) model time series data with higher $Co$ values of
$u(t)$ statistic of SMK between D4 time series and original data except for the Ishurdi station. Moreover, two
stations (Bhola and Satkhira) show significant trends in original data. The closest Z statistic is found between
original and model (D2+D4+A) time series data for both stations. Again, D4 (16-year periodicity) is the
dominant periodic component based on $Co$ for both of these stations. Therefore, 16-year periodicity is the main
periodic component which is responsible for trends in $P_{ET}$ data over the study area. Moreover, D3 (8-year)
periodicity also has some effect on the trends and present in some stations (Table 2 and also see ESM Table S1).
D4 (16-year) periodicity dominates in annual rainfall in Marmara region in Turkey (Partal and Küçük, 2006).
Araghi et al. (2016) found that 8 to 16 year (D3 to D4) periodicity is responsible for trends in annual
temperature in Iran.

### 3.2.2 Actual Evapotranspiration

All of the stations except Bogra show decreasing trends in $A_{ET}$ and the calculated Z statistic ranges from -2.90 in
Bogra station to 0.31 in Ishurdi station. Similar to the trends found in $P_{ET}$, trends in $A_{ET}$ are also insignificant at
5% significance level except Ishurdi station which shows significant (at 5% significant level) decreasing trend.

The magnitudes of the trends of original $A_{ET}$ data vary from -5 mm/year in Faridpur station to 0.75 mm/year in Bogra station. The distribution of the magnitude of the trend is shown in Figure 4b. The periodicity in $A_{ET}$ is slightly different from $P_{ET}$ (see ESM Table S2). Almost half of the (five) stations show that D2 (4-year) is the main periodic component and D4 (16-year) has also effects on trend as Z statistic of (D2+D4+A) model time series is the nearest to original series for Khulna and Ishurdi stations. Moreover, D4 (16-year) is the main periodicity for Rangpur and Rajshahi stations. In addition, D1 (2-year) is the dominant periodicity in Barisal, Bhola and Bogra stations. $A_{ET}$ depends on climatic factors such as $P_{ET}$ and rainfall as well as on soil moisture conditions. The variations in periodicity in $A_{ET}$ from $P_{ET}$, hence, are mainly related to soil moisture conditions of the area.

### 3.2.3 Surplus

Almost 82% stations show insignificant decreasing trends in annual surplus of water. The magnitude of trends of original annual surplus data ranges from -11.63 mm/year to 6.71 mm/year (Figure 4c). There is a similarity in periodicity characteristics of $P_{ET}$ and surplus (See EMS Table S3). D4 (16-year) is the main periodic component present in seven stations and in most of the cases D2 is also present (D2+D4+A) except in Rajshahi. D3 (8-year) is mainly responsible for trend in surplus in three stations. Surplus mainly occurs during the rainy season (Jun-Oct) in the study area when soil moisture is almost full and $A_{ET}$ is equal to $P_{ET}$. Surplus mainly depends on rainfall. Therefore, it also provides an idea about the periodicity in rainfall.

### 3.2.4 Deficit

Approximately 73% stations show increasing trends in the annual deficit of water. The increasing trends are significant in two stations at 95% confidence level (see ESM Table S4). However, Satkhira station shows a significant decreasing trend (Z = -2.08) in deficit. The magnitude of trends of original annual deficit data ranges from -8.1 to 7.7 mm/year (Figure 4b). The periodicity analysis reveals that D4 (16-year periodicity) is the main responsible factor for the trends in the deficit. The Z statistic of (D2+D4+A) model time series data is close to the Z statistic of original time series data (ESM Table S4). D3 (8-years periodicity) is also responsible for trends in data of two stations.

### 3.3 Model Selection and Forecasting Ability

Firstly, ARIMA model has been selected for forecasting the WBCs time series. Four-step analysis has been done during the time series modeling: (1) stationarity in the data has been checked by Augmented (ADF) test, (2) auto-correlation function (ACF) has been used for selecting the order of MA process (see ESM Fig. S2-S5), (3) partial auto-correlation function (PACF) has been used for selecting the order of AR process (see ESM Fig. S2-S5) and (4) finally, the appropriate model has been selected based on several trials, values of model selection criteria like Akaike information criterion (AIC) and Bayesian information criterion (BIC). During the trails for selecting the model, besides the manual model selection based on ACF, PACF, AIC and BIC, the auto ARIMA function of the 'forecast' package (Hyndman et al., 2017) of R (R 3.4.0 language developed by R Development Core Team, 2016) has been used to get reasonable information about the nature of the data for modeling. The best model has been selected based on lower values of AIC, BIC, and higher value of $R^2$. The Q-Q plot has been prepared to check the normality of residuals. The performance of ARIMA model (parameters can be found in ESM Table S5) has been evaluated by *NSE* and $R^2$ (Table 3). The estimated values of *NSE* of ARIMA model of

$P_{ET}$ time series vary from -0.60 for Bhola station to 0.81 for Jessore station (Table 3). ARIMA models for
almost all stations show unsatisfactory performance as the average *NSE* value of eleven stations is 0.38 and $R^2$
values range from 0.10 to 0.81 with an average of 0.38. Moreover, the *NSE* value of Bhola station indicates that
ARIMA model is not suitable for forecasting the $P_{ET}$. ARIMA model has also been applied to $A_{ET}$, surplus and
deficit time series data. After carefully checking the ACF and PACF (see ESM Figure S2–S5) of $A_{ET}$, it is found
that there are no significant spikes in ACF and PACF. Moreover, the results obtained from auto ARIMA
functions also show similar results. Therefore, ARIMA model is not satisfactory for forecasting the variability
or changing pattern of $A_{ET}$. For WBCs like surplus and deficit, the performance of ARIMA model is almost
similar to $A_{ET}$ except for few cases. As the hydro-meteorological data are affected by noises from different
hydro-physical processes (Wang et al., 2014), results obtained from ARIMA models show the unsatisfactory
performance. To improve the model performance, it is necessary to remove the noise from the data. DWT
denoising has been applied to the WBCs data in the present study and the quality of the denoising time series
data has been checked before further processing. The important criteria to select a method for denoising the time
series using wavelet transformation are the mean of the original series and denoising time series data should be
close and standard deviation of denoising time series should be less than the original series (Wang et al., 2014).
Figure 5(a) displays mean of the actual time series of $P_{ET}$ and mean of wavelet denoising time series of $P_{ET}$. It is
seen that there are no visible differences between the mean of the original time series data and DWT wavelet
denoise time series data. Moreover, the standard deviation of $P_{ET}$ of wavelet denoising time series is lower than
the original time series (Figure 5b). $A_{ET}$, surplus and deficit time series also show the similar results (see ESM
Figure S4–S5). Furthermore, lag-1 auto-correlation of wavelet denoise time series data must be higher than the
original time series (Wang et al., 2014). For this consideration, wavelet denoise time series also shows that lag-1
absolute value of auto-correlation is higher than that of original series value [see ESM Figure S2 (b), S3 (b), S4
(b) and S5 (b)]. The performance of WD-ARIMA model is shown in Table 3. After denoising the data, the
performance of ARIMA model is satisfactory for all WBCs time series data (Table 3). The average *NSE* value
of WD-ARIMA models for $P_{ET}$ time series of eleven stations located in the western part of Bangladesh is 0.76
and an average $R^2$ value is 0.67. Both performance indicators reveal that the performance of the WD-ARIMA
model is better than the classical ARIMA model (Table 3). Moreover, the average *NSE* value of WD-ARIMA
models of $P_{ET}$ time series of these stations is 0.92 which indicates that the performance of the model is very
good and the average $R^2$ value is 0.89 which indicates the model can explain almost 89% variance of the data
(Table 3). Results obtained from WD-ARIMA models of annual surplus and annual deficit also indicate very
good performance for forecasting these variables (Table 3). The average *NSE* value of eleven stations of WD-
ARIMA models for the annual surplus is about 0.92 and average $R^2$ value is 0.90. WD-ARIMA models for
forecasting the annual deficit (average *NSE* = 0.88) also show good performance. The comparative study of the
performance of the WD-ARIMA models of WBCs reveals that model performance is very good or good for $A_{ET}$,
annual surplus and deficit. However, the performance is acceptable for $P_{ET}$. This deviation may arise from the
variability of the $P_{ET}$ is higher than others WBCs or may relate to the variability of climatic variables.
Moreover, validations of the models have been done to explore the forecasting ability of the fitted models. The
mean percentage error ($E_{MP}$) of the forecasted values for the four year period from 2008-09 to 2012-13 has been
calculated to know the percentage bias of the forecasted data (Table 4). The average $E_{MP}$ of eleven stations of
WD-ARIMA models for $P_{ET}$ is -0.6 (with ranges from 0.75 to -3.34) that indicates the forecasted values are
slightly lower than the actual values. The typical plots of the actual time series data versus fitted model data, the
normal Q-Q plot of residuals of the models, and actual and observed values of WBCs (plots for all stations can
be found in ESM Fig. S6–S9) are shown in Figure 6. The plot of actual versus forecasted values (Figure 6)
indicates that generally the actual versus forecasted values are very close for the hydrologic years 2009-10 and
2010-11. However, the differences are generally increasing after these periods for all WBCs (also see ESM
Figure S5). Moreover, the actual versus the model calculated fitted values are very close to each other. The
normal Q-Q plots reveal that the residuals of the models are near normal. The $E_{MP}$ values of WD- ARIMA
models for $A_{ET}$ range from -0.7 to 0.2 with an average of -0.09 which also indicates that forecasted $A_{ET}$ values
are slightly lower than actual $A_{ET}$ values. The $E_{MP}$ values for annual surplus (average = -0.75) and annual deficit
(average = -0.12) are almost similar to the $A_{ET}$ and $P_{ET}$. It is also notable that the average $E_{MP}$ values for all
WBCs are negative, which indicate the forecasted values of WBCs are slightly lower than the actual values for
most of the stations.

## 3.4 Discussion

The present study reveals that a decreasing trend in $P_{ET}$ dominates over the study area. However, positive trends
in rainfall and temperature dominate in the western part of Bangladesh (e.g. Shahid and Khairulmaini, 2009;
Kamruzzaman et al., 2016a). Moreover, a recent study has also found a negative trend in evapotranspiration in
four stations located in northwest Bangladesh (Acharjee et al., 2017). Though annual rainfall and temperature of
Satkhira station show positive trends (Kamruzzaman et al., 2016a), $P_{ET}$ shows a significant downward trend.
Increasing trends in temperature have been found in Yunnan Province of South China, but $P_{ET}$ shows decreasing
trend (Fan and Thomas, 2012). McVicar et al. (2012) have also found decreasing trends in $P_{ET}$ in the different
parts of the world. Therefore, temperature-based models for the estimation of $P_{ET}$ cannot well explain the causes
of changes in $P_{ET}$, though the temperature is the primary driver of changes in $P_{ET}$ (IPCC, 2007). To get a
detailed idea about the underlying mechanisms of changes in $P_{ET}$, it is necessary to do a detailed analysis of all
climatic variables such as rainfall, temperature, sunshine hours, wind speed, humidity and climate controlling
phenomena like El Niño Southern Oscillations (ENSO).
The study has also developed WD-ARIMA models for forecasting the WBCs. The performance of the model
shows the benefit of denoising of hydrological time series data like $P_{ET}$, $A_{ET}$, surplus and deficit. However, the
model performance analysis criterion like *NSE* indicates that the performance of the model for $P_{ET}$ forecasting is
acceptable ($NSE \geq 0.65$). To have a closer look at the forecasted values and actual values, the deviation between
forecast values and actual values increases with increasing time steps. Therefore, WD-ARIMA models are not
suitable for long-term forecasting. The present study has developed the WD-ARIMA model by coupling the
discrete wavelet denoise time series data and ARIMA model. The soft threshold method has been selected for
denoising the time series data and universal threshold (UT) method which has been used for the determination
of the threshold value. However, there are some approaches for threshold value determination such as SURE
(Stein, 1981), MINMAX (Donoho and Johnstone, 1998) and so on. Moreover, Wang et al. (2014) develop a
hybrid approach for denoising the hydro-meteorological time series such as rainfall and streamflow called
adaptive wavelet de-noising approach using sample entropy (AWDA-SE). The study has shown that the
performance of the developed denoising method is better than conventional de-noising methods for denoising
rainfall and streamflow. These approaches may apply to increase the performance of ARIMA models for
forecasting hydrological variables like $P_{ET}$. Moreover, there are several mother wavelet families such as
Daubechies, Harr, Coiflets, Morlet, Mexican Hat and so on (Sang, 2013). In the present study, only Daubechies-
6 from Daubechies wavelet family has been applied as mother wavelet of discrete wavelet transformation. WD-
ARIMA models for forecasting the $A_{ET}$, surplus and deficit show very good performance, whereas the classical
ARIMA model shows poor performance or unable to forecast the WBCs. Moreover, studies (e.g. Chou, 2011;
Kisi, 2008; Partla, 2009; Santos and da Silva, 2014; Rahman and Hasan, 2014; Nury et al., 2016; Adamowski
and Chan, 2011; Khalek and Ali, 2016) have also mentioned that the performance of wavelet aided models for
forecasting non-stationary hydro-meteorological variables is better than classical ARIMA and ANN models. As
the traditional methods such as Wiener filtering, Kalman filtering, Fourier transform are not suitable for non-
stationary hydrological time series data (Adamowski and Chan, 2011; Sang, 2013), wavelet denoising can be
used to improve the performance of the classical ARIMA models for forecasting hydrological variables.
## 5. Summary and Conclusions
The study explores the changes in WBCs using wavelet aided various forms of MK test and develops wavelet
aided ARIMA models for forecasting the WBCs. The results obtained from trends analysis indicate that
decreasing trends are dominant in all WBCs in the western part of Bangladesh during the period of 1982-83 to
2012-13. However, most of the trends are insignificant at 95% confidence level. One positive and one negative
significant trend in $P_{ET}$ have been found in Satkhira and Bhola stations respectively. The study analyzed
different combinations of D and A (i.e. D+A and D+A+A) components of DWT with $Co$ of $u(t)$ statistic of
SMK test that provides details information about the dominant periodicity that clearly affects the trend in
original data and the time period which has also effect on trend in data (see section trend and periodicity or for
example of Bhola station). The findings of the study reveal that to get details about the time period responsible
for trends in data, it is necessary to analyze different combinations of D+A and D+A+A components rather than
only details component (D) or approximation of wavelet transform data. Moreover, the study explored that
changes in temperature or rainfall or both of these are not only associated with changes in $P_{ET}$. Before
concluding the attribute of changes in $P_{ET}$, it is necessary to do details analysis of all the relevant climatic
variables. In the western part of Bangladesh, D3 (8-year) and D4 (16-year) components have dominant effects
on trends in original WBCs time series data. D2 (4-year) periodicity are also present in some cases, especially
for $A_{ET}$. As surplus occurs during the rainy season and most of the rainfall occurs during this season, it may
point out that rainfall pattern may have a similar periodicity (D3 to D4).
Modeling of the study reveals that WBCs time series data is affected by noises from different hydro-physical
interactions. As a result, classic ARIMA models show unsatisfactory performance for most of the cases (for
example $P_{ET}$) or unable to model the variability and changes in $A_{ET}$, surplus and deficit. The study has showed
that ARIMA model can be used to model the WBCs time series after the denoising the WBCs time series using
DWT with a universal threshold. The quality of wavelet denoise time series data has been evaluated and found
satisfactory results for WBCs denoising. The fitted WD-ARIMA model performance has been evaluated by $NSE$
and $R^2$ (average $NSE$ and $R^2$ values of eleven stations located in western part of Bangladesh are 0.76 and 0.67
for $P_{ET}$; 0.92 and 0.89 for $A_{ET}$; 0.92 and 0.90 for annual surplus, and 0.88 and 0.88 for annual deficit
respectively). The validation of WD-ARIMA models shows acceptable to very good performance for the short-
term forecasting of WBCs as the validation for the period of 2009-10 to 2012-13 shows the acceptable $E_{MP}$
value. However, the gap between the actual data and forecasted data increases with increasing time period. The
obtained results are encouraging for further studies to find out a realistic model for real-world application under
the changing climate. The results of the study can be incorporated into water resources management plans for
highly irrigated western part of Bangladesh where groundwater resource is at a critical stage. Further studies,
therefore, denoising of hydrological time series data using different mother wavelets such as Haar, Coiflet and
determination of thresholds using MINMAX, SURE or entropy based adaptive denoising approaches would be
helpful for developing the better models for hydro-climatic time series in the context of climate change and
would be beneficial for managing water resources in a sustainable manner.

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

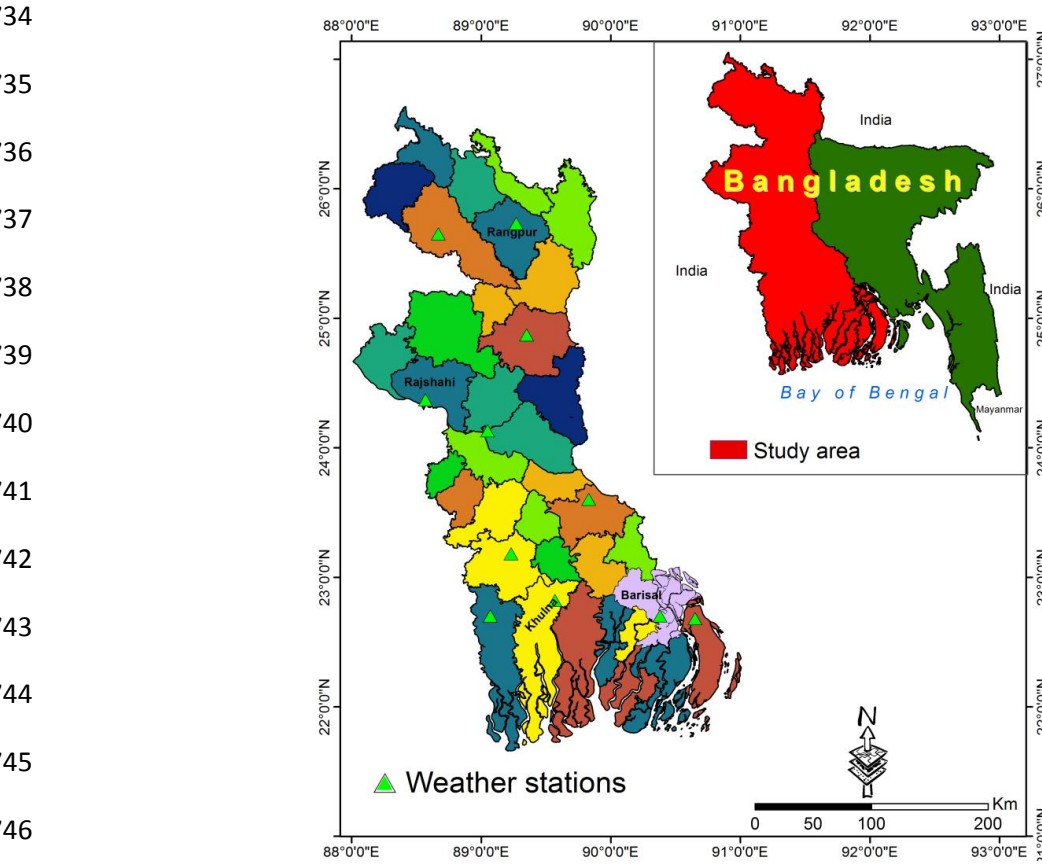


**Figure 1: Study area western part of Bangladesh with locations of meteorological stations.**








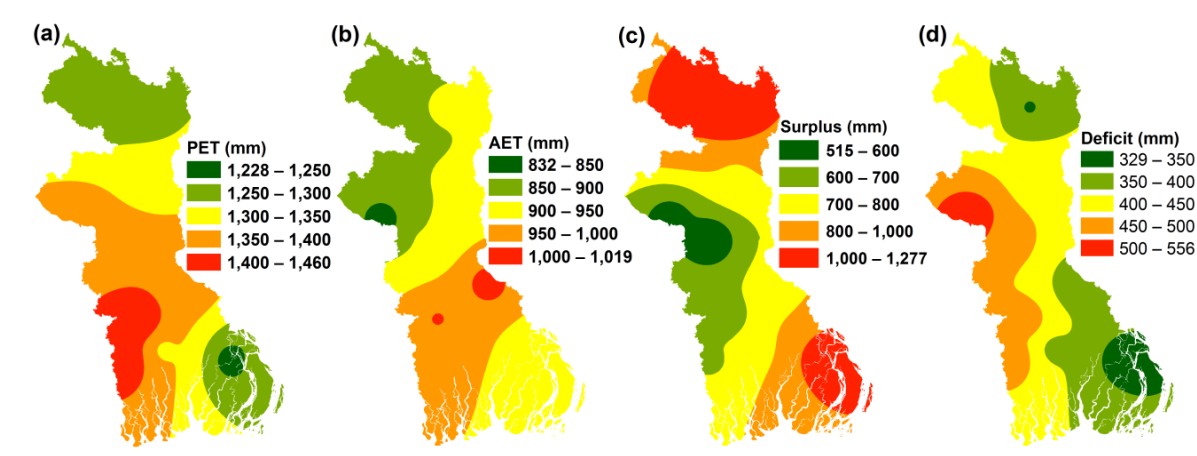



**Figure 2: Distribution of mean annual (a) $P_{ET}$, (b) $A_{ET}$, (c) surplus and (d) deficit of water in the study**
**area during the hydrologic year 1981-82 to 2012-13.**

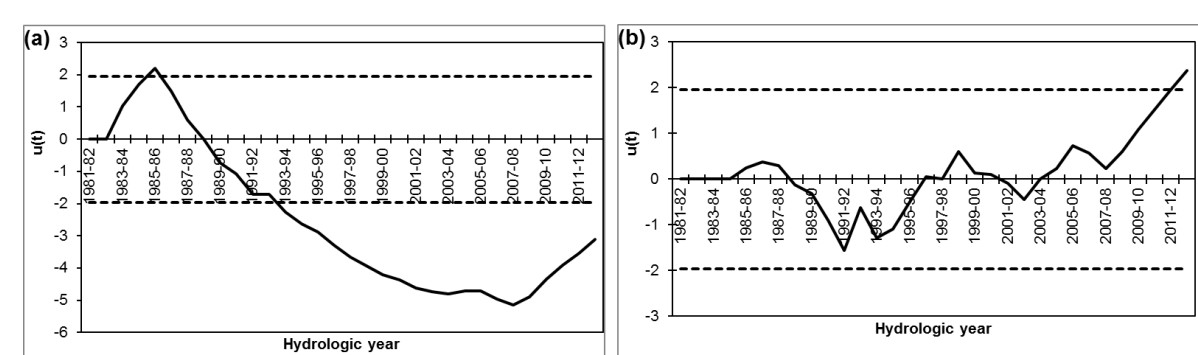

**Figure 3: Sequential values of the statistics *u (t)* of (a) Satkhira station and (b) Bhola station.**

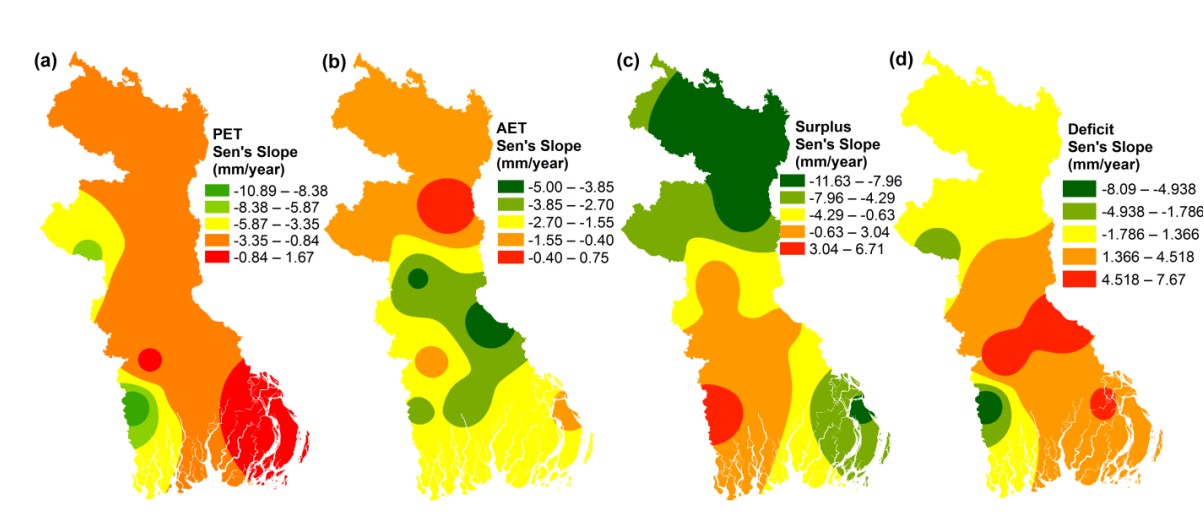

**Figure 4: Distribution of rate of changes of WBCs during the period of 1981-82 to 2012-13.**

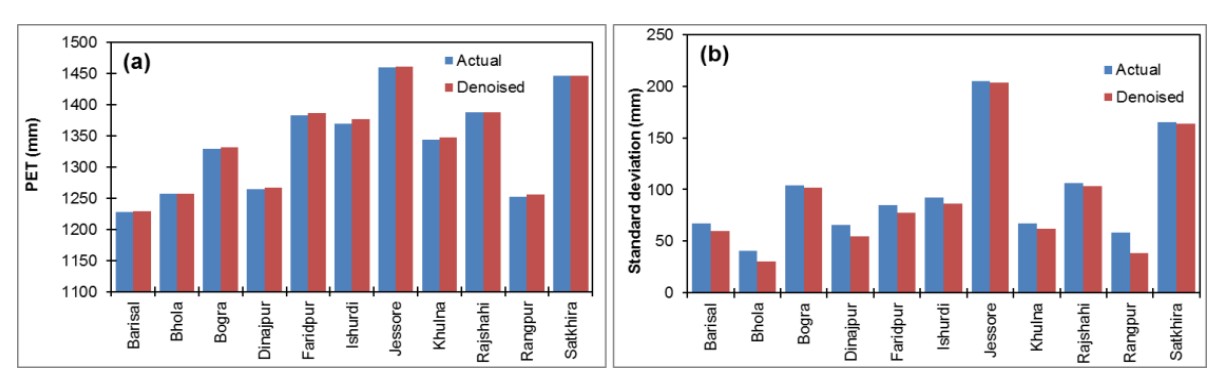

**Figure 5: Comparison between actual and wavelet denoise $P_{ET}$ time series (a) mean and (b) standard deviation.**

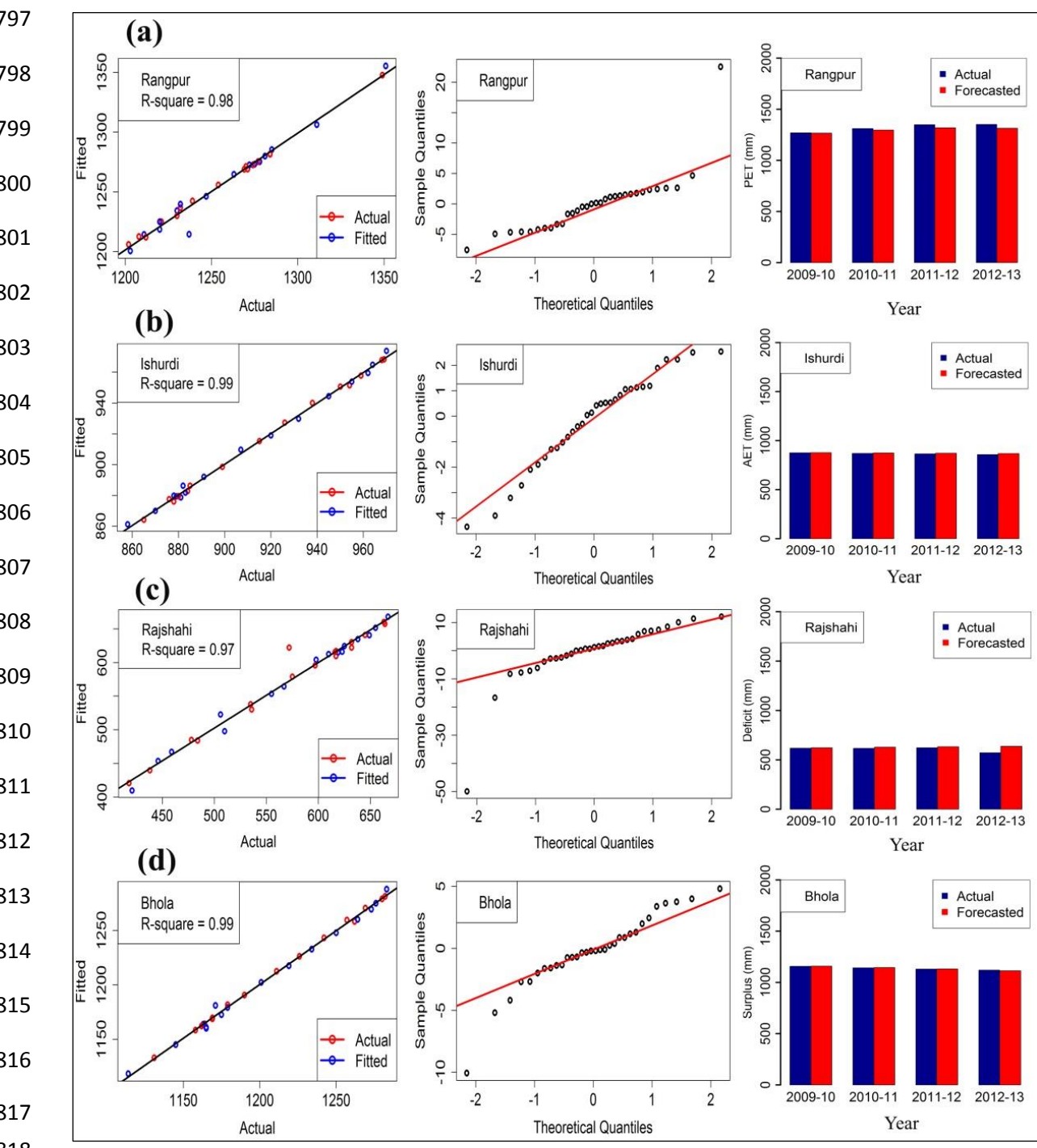

**Figure 6: Plot of best WD-ARIMA model first panel represents actual versus fitted values for the period of 1981-82 to 2012-2013, the second panel is normal Q-Q plot of residuals of the model, and the third panel shows actual, fitted and forecasted values for 2009-2010 to 2012-13 (a) $P_{ET}$ of Rangpur station located in north; (b) $A_{ET}$ of Ishurdi station located in the central part, (c) deficit of Rajshahi station located in NW Bangladesh and (d) surplus of Bhola station located in south of the study area.**

**Table 1**: Calculations of water balance components (Thornthwaite and Mather, 1957)

|  | Wet months $(P - R_0) > P_{ET}$ | Dry months $(P - R_0) < P_{ET}$ |
|---|---|---|
| $A_{ET}$ | $P_{ET}$ | $(P - R_0) + \Delta S_B$ |
| Deficit | 0 | $P_{ET} - A_{ET}$ |
| Surplus | $(P - R_0) - P_{ET}$ | 0 |

Where $P$ is the rainfall (mm), $R_0$ is the direct runoff (mm), $P_{ET}$ is the potential evapotranspiration (mm), $A_{ET}$ is the actual evapotranspiration (mm) and $\Delta S_B$ is the changes in soil moisture storage (mm).

**Table 2:** Z statistic of MK or MMK of original time series, approximation and different models $P_{ET}$ of DWT (the dominant components are bold and asterisk for significant at 5% level)

| Stations / Models | Barisal | | | Bhola | | | Bogra | | | Dinajpur | | |
|---|---|---|---|---|---|---|---|---|---|---|---|---|
|  | *Z* | *Co* | *MSE* | *Z* | *Co* | *MSE* | *Z* | *Co* | *MSE* | *Z* | *Co* | *MSE* |
| Original | 0.72 | | | 2.37* | | | -0.20 | | | -0.98 | | |
| A | -1.80 | 0.24 | 11.56 | -1.80 | -0.15 | 17.15 | -1.80 | 0.83 | 4.66 | -1.80 | 0.83 | 3.47 |
| D1 | 0.91 | 0.50 | 0.50 | 2.02* | 0.25 | 0.68 | 1.16 | -0.42 | 5.10 | - | | |
| D2 | -0.03 | 0.17 | 1.51 | 0.61 | 0.21 | 0.94 | 0.16 | 0.60 | 3.70 | 0.43 | 0.63 | 8.82 |
| D3 | 0.45 | 0.17 | 1.51 | 0.46 | 0.21 | 0.94 | 1.08 | 0.60 | 3.70 | 0.90 | 0.63 | 8.82 |
| D4 | **0.76** | 0.37 | 3.93 | 1.20 | 0.80 | 7.28 | 1.14 | 0.13 | 3.76 | 2.10* | -0.03 | 13.35 |
| D1+A | -0.89 | 0.35 | 0.71 | 1.58 | 0.11 | 0.72 | -2.35* | 0.90 | 0.54 | -1.70 | 0.95 | 0.44 |
| D2+A | -1.51 | 0.14 | 2.75 | 0.48 | 0.13 | 1.05 | -1.54 | 0.89 | 0.62 | -2.05* | 0.93 | 1.25 |
| D3+A | -0.66 | 0.50 | 1.90 | 0.31 | 0.14 | 1.23 | -1.91 | 0.89 | 5.72 | -1.56 | 0.95 | 3.03 |
| D4+A | 0.06 | 0.53 | 9.99 | 0.90 | 0.77 | 8.71 | -0.34 | 0.58 | 7.32 | -1.79 | 0.85 | 2.41 |
| D1+D2+A | -0.89 | 0.35 | 0.82 | 0.73 | 0.39 | 0.68 | -1.12 | 0.88 | 0.77 | -1.76 | 0.97 | 0.18 |
| D1+D3+A | -0.81 | 0.58 | 0.88 | 0.79 | 0.31 | 0.69 | -1.33 | 0.87 | 0.89 | -1.51 | 0.98 | 0.38 |
| D1+D4+A | 0.91 | 0.63 | 1.16 | 2.29* | 0.83 | 0.35 | 0.24 | 0.87 | 0.53 | -1.15 | 0.97 | 0.20 |
| D2+D3+A | -0.46 | 0.43 | 1.24 | 1.01 | 0.08 | 2.42 | -1.33 | 0.89 | 1.10 | -1.37 | 0.96 | 1.35 |
| D2+D4+A | 0.54 | 0.50 | 2.84 | **2.36*** | 0.77 | 0.68 | 0.10 | 0.88 | 0.60 | **-1.27** | 0.94 | 0.85 |
| D3+D4+A | 0.56 | 0.85 | 2.04 | 1.83 | 0.90 | 0.74 | **-0.30** | 0.87 | 1.37 | -1.54 | 0.96 | 2.10 |

*MSE*, total mean square error; *Co*, correlation between original data and DWT models

**Table 3:** Comparison of performance of ARIMA model and WD-ARIMA model

| Stations | $P_{ET}$ | | | | $A_{ET}$ | | Surplus | | Deficit | |
|---|---|---|---|---|---|---|---|---|---|---|
| | ARIMA | | WD-ARIMA | | WD-ARIMA | | WD-ARIMA | | WD-ARIMA | |
| | *NSE* | $R^2$ | *NSE* | $R^2$ | *NSE* | $R^2$ | *NSE* | $R^2$ | *NSE* | $R^2$ |
| Barisal | 0.42 | 0.43 | 0.95 | 0.57 | 0.58 | 0.58 | 0.99 | 0.99 | 0.87 | 0.87 |
| Bhola | -0.57 | 0.10 | 0.95 | 0.61 | 0.98 | 0.59 | 0.99 | 0.99 | 0.56 | 0.67 |
| Bogra | 0.52 | 0.50 | 0.68 | 0.63 | 0.97 | 0.97 | 0.99 | 0.99 | 0.95 | 0.95 |
| Dinajpur | 0.54 | 0.52 | 0.99 | 0.79 | 0.98 | 0.98 | 0.84 | 0.95 | 0.95 | 0.94 |
| Faridpur | 0.32 | 0.30 | 0.65 | 0.50 | 0.99 | 0.99 | 0.99 | 0.99 | 0.87 | 0.88 |
| Ishurdi | 0.34 | 0.31 | 0.39 | 0.57 | 0.99 | 0.99 | 0.98 | 0.56 | 0.88 | 0.89 |
| Jessore | 0.81 | 0.81 | 0.76 | 0.67 | 0.82 | 0.82 | 0.96 | 0.96 | 0.82 | 0.77 |
| Khulna | 0.31 | 0.29 | 0.45 | 0.41 | 0.98 | 0.97 | 0.99 | 0.99 | 0.94 | 0.94 |
| Rajshahi | 0.58 | 0.56 | 0.60 | 0.61 | 0.99 | 0.99 | 0.98 | 0.98 | 0.97 | 0.97 |
| Rangpur | 0.19 | 0.20 | 0.98 | 0.98 | 0.84 | 0.92 | 0.47 | 0.49 | 0.86 | 0.84 |
| Satkhira | 0.77 | 0.20 | 0.95 | 0.98 | 0.99 | 0.99 | 0.99 | 0.99 | 0.99 | 0.99 |
| **Avg.** | **0.38** | **0.38** | **0.76** | **0.67** | **0.92** | **0.89** | **0.92** | **0.90** | **0.88** | **0.88** |



**Table 4:** Accuracy of WD-ARIMA models of WBCs for validation of the model's predictive ability for the
period of 2009-10 to 2012-2013

| Stations | $P_{ET}$ | | $A_{ET}$ | | Surplus | | Deficit | |
|---|---|---|---|---|---|---|---|---|
| | $E_M$ | $E_{MP}$ | $E_M$ | $E_{MP}$ | $E_M$ | $E_{MP}$ | $E_M$ | $E_{MP}$ |
| Barisal | 0.07 | -0.02 | -5.36 | -0.70 | -0.70 | -0.10 | 0.80 | 0.29 |
| Bhola | 0.75 | 0.06 | -0.10 | -0.01 | -0.80 | -0.10 | 0.80 | 0.29 |
| Bogra | -0.75 | -0.19 | 0.19 | 0.02 | -1.10 | -0.10 | -0.07 | -0.03 |
| Dinajpur | -0.16 | -0.01 | -0.19 | -0.02 | -0.10 | 0.00 | -0.17 | -0.10 |
| Faridpur | -2.22 | -0.25 | -0.77 | -0.07 | -0.10 | 0.00 | 1.05 | 0.39 |
| Ishurdi | 0.34 | -0.16 | -0.45 | -0.05 | -0.20 | 0.00 | 0.72 | 0.25 |
| Jessore | 0.11 | -0.02 | 0.26 | 0.02 | 0.70 | 0.00 | 1.52 | -2.42 |
| Khulna | -1.56 | -0.22 | -0.53 | -0.05 | 0.60 | 0.10 | 0.01 | -0.01 |
| Rajshahi | -3.34 | -0.35 | -0.11 | -0.01 | -0.60 | -0.10 | -0.14 | 0.08 |
| Rangpur | -0.11 | -0.01 | -0.40 | -0.05 | -8.50 | -7.90 | -0.05 | -0.14 |
| Satkhira | 0.54 | 0.04 | -0.36 | -0.04 | 0.50 | 0.10 | -0.43 | 0.12 |
| **Avg.** | **-0.57** | **-0.10** | **-0.71** | **-0.09** | **-0.95** | **-0.75** | **0.37** | **-0.12** |






