# Peer review of "Modeling the changes in water balance components of highly irrigated western part of Bangladesh"

_Hydrology and Earth System Sciences, 2017_

## Referee Comment (RC1) · Anonymous Referee #1 · 14 Nov 2017

Overall Comments: This paper by A.T.M. Sakiur Rahman et al (the authors) describes the development of wavelet autoregressive moving average models to forecast changes in water balance components. The approach is applied to a highly-irrigated area in Western Bangladesh using data collected between 1982 and 2013. The authors show that the approach can be used to forecast short term changes in water balance components but suggest that models could be further improved using different combinations of wavelet analysis.

Overall, the paper is well structured, although the level of English used, and the wording of some sentences could be refined. This paper fits well within the scope of HESS

[Figure]

Creative Commons BY license logo

and the described methodology provides a useful approach to investigating changes in water balance. I would recommend this paper is published, subject to some minor corrections, including improvement in the quality of English used, along with consideration of the elements described below.

Specific Comments: This is a technical paper with a strong methodological focus. While the approaches used are well described, I would recommend the authors more clearly define the importance of the work in a broader hydrological context; for example, the relevance to hydrology and water resource management regionally and globally as this is only mentioned briefly.

This study is applied to a region of Bangladesh, however, both the approach and the paper would benefit if the transferability of the methodology could be highlighted by the authors; i.e. in what environments and under what conditions would the methods described work well.

The paper would also benefit from a separate limitations section; for example, as an additional section 3.4. Limitations, in addition to those found within the methods, should also be highlighted. This may include, for example, scarcity of data. Also, data from 11 stations are used to represent over 61,000 km2. A comment on how representative this data is would be welcome.

Some of the text in Figure 6 is difficult to read. Also, please be consistent with the font.

Some examples of typos and where sentence rephrasing would be beneficial: The following list is by no means exhaustive and the paper would greatly benefit from a detailed proof read before re-submission.

Line 53: "Almost most of the studies found. . .." Please rephrase.

Line 71/72: "Water balance study has not attracted much attention in Bangladesh too." Please rephrase.

Line 144: "The reaming amount of rainfall has been applied to calculation". Possibly

change or rephrase to "The remaining amount of rainfall has been included in the calculation".

Line 430: "...it is necessary to do detailed analysis...".

Line 437: "...as a result...".

---

## Referee Comment (RC2) · Anonymous Referee #2 · 15 Nov 2017

This paper investigates changes in water balance components between 1981-82 to 2012-13 in an area of Bangladesh that is intensively irrigated. First, historical trends are examined using the Mann-Kendal test and discrete wavelet transformation. Then, ARIMA models are developed in order to forecast changes in water balance components. The paper produces some interesting results; particularly around the use of ARIMA models that are fitted to wavelet denoised time series data.

The paper is well organised and about the right length for a study of this kind. Generally speaking the equations are well laid out and easy to follow. However, as it stands the level of English used in the paper is poor, which makes some sections very difficult

to follow. I would strongly advise the authors to consult a proofreader who has full professional proficiency in written English. Nevertheless in my judgement the scientific content is sound and represents an interesting approach to analysing and forecasting changes in the water balance. Thus, I would reconsider this paper for publication following a major revision to improve the quality of English as well as addressing the specific points mentioned below.

1. The Introduction lacks focus, does not provide much critical analysis and does not place the work in the broader context of water resources management. I would like to see the aim of the paper clearly stated in the first paragraph of the paper, so that readers know what the paper is setting out to achieve. The paper is methodological in nature, so the Introduction must make clear to the reader the state-of-the-art in time series analysis for water resources management. Line 82 onwards does include some critical analysis but, in my opinion, it is insufficient to persuade the reader of the approach and its relevance to hydrology more generally. Focus less on the results of studies and instead examine and compare the different ways that previous researchers have tackled the problem.

2. The first half of Section 2.3.1 is probably superfluous: it is well known that Penman-Monteith is the most appropriate method to use to calculate PET, data permitting.

3. Line 137: I'm not exactly sure what 'Deficit' and 'Surplus' mean in this context (nor is it clear why they are capitalised) – provide additional explanation.

4. Line 137-139: It is presented as a fact that 'the concept of water balance in unsaturated zone...give the best estimation for the real world' - this is quite a statement and surely unjustified. I note that Bakundukize et al 2011 were investigating hydrology in Burundi – are there similarities to Bangladesh? Provide some additional arguments for using the Thornthwaite and Mather model.

5. Line 143: Wolock and McCabe (1999) examined hydrology in the United States – is it reasonable to assume a 5% runoff in Bangladesh, given its tropical climate?

6. Line 145-151: Express the water balance model as equations. The calculation of the water balance is fundamental to the subsequent analysis, so it should be clear what you have done.

7. In my view Section 2 would benefit from a short overview describing the reason for carrying out the various steps (water balance, Mann-Kendal, wavelet analysis, ARIMA) and how they relate to one another. At the moment this is not clear.

8. Line 154: Which hydrological variables were investigated?

9. Line 159: What are these 'Z' values, and what is their importance? This is the first time they have been mentioned.

10. Section 2.3.7 is unnecessary here unless it specifically influences the scientific results. Instead put this information in an Appendix or similar (however, I congratulate the authors for putting their computer code alongside the paper – this is not done often enough).

11. I think Section 3 should simply describe the results, with an additional 'Discussion' section for placing the results in the context of other studies (e.g. Line 281, 288, 321 etc. should be put in a Discussion section). The discussion should include additional analysis discussing the various limitations and weaknesses of the present study as well as suggesting improvements.

12. Line 265: Use 'Potential evapotranspiration' rather than 'Pet' in section headings.

13. Line 359: This is just a piece of computer code – what does it do, and what insight does it provide that you cannot gain from manual interpretation of ACF, PACF, AIC, BIC?

14. Line 362: What is a Q-Q plot?

15. Line 386-416: To my mind this passage is the strongest part of the paper – the discussion should emphasise this result and its relevance to water resources management more generally.

16. As I mentioned earlier, I would strongly suggest creating an additional Discussion section in which to discuss the results in the context of other studies, highlight limitations and propose future research directions.

---

## Short Comment (SC1) · 4 Dec 2017

This manuscript describes the application of discrete wavelet transformation (DWT) and different forms of Mann-Kendal test to study changes in water balance components (WBCs). The authors also develop a "wavelet autoregressive moving average (ARIMA) model" to forecast WBCs. The contribution of the manuscript seems to be detecting trends and identifying periodicities in WBCs along with forecasting them after removing the noise from the time series. The manuscript is statistical than hydrological and I would say that hydrological concepts are insufficiently addressed and not fully developed. Moreover, there are some theoretical inaccuracies and confusing statements

(especially in hydrologic side) that undermine the quality of this manuscript. Overall, the authors address an interesting subject; but in the current form, there are concerns and shortcomings that warrant major revisions.

Below I summarize some constructive suggestions for improvement:

Theoretical issues:

1- There is a confusion in the paper about the concept of "Water Balance" and its "Components". Water Balance Components (WBC) and some other parameters are frequently used in awkward and confusing sentences. As an example, potential evapotranspiration (PET), which is named as one of the WBCs in line 91 is called "the key parameter to estimate water balance components ..." in line 119. I strongly recommend that the authors provide the Water Balance Equation, briefly introduce Water Balance Components, and define which components they consider in their study, clearly. They may explain these concepts at the beginning of the "Methods" section (section 2.3). They may also mention the reason(s) for selecting each WBC.

2- Following the previous comment, both Potential Evapotranspiration (PET) and Actual Evapotranspiration (AET) are considered, surprisingly, as components of water balance equation (for example in lines 91, 265, 325) without any explanation on their application and role in the equation. However, the application of these parameters in Water Balance Equation is different and they cannot be considered both at the same time. I would also suggest the authors revise their manuscript to ensure that no confusing sentence remains on this subject.

3- In line 145, the authors stated that "When rainfall is greater than PET the soil always remains full of water and . . .", which is an inaccurate statement. I understand the authors try to explain the concept of surplus; however, surplus occurs when the soil becomes saturated and infiltration is hardly possible.

Title and Abstract:

4- The authors should perhaps reframe the title to better reflect their work. The present title implies that the study is mainly concentrated on the interaction between changes in water balance components and intensive irrigation in Western Bangladesh.

5- I would recommend that the authors name water balance components that they consider in this study in the abstract. They may provide then a summary of the methodology and results in a more organized way.

6- I was wondering whether the authors apply ARIMA or ARMA models in their study. In case of having ARIMA, which stands for "Autoregressive Integrated Moving Average" they should revise the statement in line 17.

7- Line 34: The statement ". . . findings of study can be used to improve water resources management . . ." is too generic. Please clarify in what respect this study can improve water resources management in the highly irrigated area.

The Structure:

8- In general, the paper has no flow and each section seems to be a separate part without proper connection to the other sections. I think the authors should improve the structure and flow of their manuscript.

9- I believe that the "Introduction" should significantly revised. For instance, the literature review on periodicity and using wavelet transformation is only limited to few sentences. The authors can elaborate more on what the previous researchers have done and how this study differs from previous attempts.

10- In section 2.2, "Data", I would suggest that authors provide the time duration they used in this study.

11- Headings are awkward and in some cases poorly selected. For example, in lines 265 and 325, (sections 3.2.1, and 3.2.2) it would be better to replace "PET" with "Potential Evapotranspiration" and "AET" with "Actual Evapotranspiration" respectively.

12- In section 2.3, "Methods", I would suggest that the authors provide a general overview of their methods and then explain each section in detail rather than starting the section immediately with a sub-heading.

13- Section 3, which seems to provide results of the study, is poorly structured. Sentences are awkward and poorly written, which makes it difficult for readers to follow.

14- I was wondering why the authors consider the section "Model Selection and Forecasting Ability" as a sub-heading of Results section (line 352). The methodology of the modeling and considerations regarding model selection should be discussed in the "Methods" section.

15- Following the above-mentioned comment, section 3.3 (lines 352-416) contains the model selection, methodology, results, and some discussions. The section is too long and without proper flow. I suggest the authors break this section into methodology, results, and discussion to help readers better follow their work.

16- The authors use passive voice and active sentences alternatively in the manuscript. They may re-write these complicated parts. For example lines 293-294.

17- In the "Summary and Conclusion", the authors mostly repeat some parts of the manuscript. I would expect to read a more conclusive summary and conclusion. For example, in Lines 447-449, (as mentioned earlier in comments on the Abstract) the authors stated that results of this study "can be incorporated to water resources management plans . . ."; but they didn't explain how this incorporation would take place. I suggest that authors add some explanations to the manuscript to clarify in what respect their work will affect water resources management in the highly irrigated lands.

Other Comments:

18- Lines 43-44: confusing and awkward statement: "Two important climatic variables like rainfall and PET that derives from the climatic variables are the main inputs in the water balance modeling". Please re-write the sentence.

19- Lines 74-77: Please re-write the statement.

20- Lines 81-82: "... most of studies were limited to detect trends or forecasting of rainfall and temperature and few studies on PET and water balance." References are required.

21- In section 2.1, it is stated that rice, the main crop cultivated in Bangladesh is mainly rain-fed or irrigated by groundwater resources (lines 104-106). Unfortunately, the authors have not clearly explained the relation between their study and irrigated area or even irrigation water demand in the study area. They may define how their work will affect the "Highly Irrigated Western Part of Bangladesh".

22- Lines 144-147, as acknowledged earlier, the statement needs theoretical revision. However, references are required for the definitions of surplus and deficit.

23- For the statement in lines 147-151, on the AET and its "calculation", references are required.

24- Lines 398-400, awkward sentence. Please re-write this sentence.

25- In general, the writing can be significantly improved. The manuscript suffers from several poorly written sentences, awkward expressions, and some grammatical errors. Some of the sentences in need of being re-written are mentioned earlier. Some other examples include:

a. Line 34: "... findings of study ..."

b. Line 35: "... in highly irrigated ..."

c. Line 48: "... attracted attention for Bangladesh."

d. Line 53: "Almost most of the studies..."

e. Line 408: "...verses...". This sentence is also long and confusing. The authors may re-write this statement.

Overall, the subject of this manuscript is interesting, and of relevance to HESS readership. Therefore, following major revisions, some of which are mentioned above, it has the potential to turn into a good publication.

---

## Author Comment (AC1) · 2 Jan 2018

We thank the referee (#1) for reviewing our manuscript entitled 'Modeling the Changes in Water Balance Components of Highly Irrigated Western Part of Bangladesh' and for his/ her valuable comments to improve our manuscript. We have responded to referee (#1) comments below**:**

**Overall Comments:**

This paper by A.T.M. Sakiur Rahman et al (the authors) describes the development of wavelet autoregressive moving average models to forecast changes in water balance components. The approach is applied to a highly-irrigated area in Western Bangladesh using data collected between 1982 and 2013. The authors show that the approach can be used to forecast short term changes in water balance components but suggest that models could be further improved using different combinations of wavelet analysis.

**Reply to overall comments:**

*We are very much grateful to you for your valuable comments about our study **'Modeling the Changes in Water Balance Components of Highly Irrigated Western Part of Bangladesh'**. We have gone through your comments and we will incorporate the necessary corrections in the relevant sections. We are also doing necessary corrections in language which is your main concern regarding the manuscript. Thank you very much for your suggestions that will help us to prepare a good paper. Yes, the paper is methodological in nature; we have tried to forecast water balance components (WBCs) more preciously after denoising the time series by discrete wavelet transformation. We also expect that there is a scope for further improvement of the methodology by different combinations of wavelet techniques. We will add a discussion section (3.4) where we discuss the advantages and limitations of our study. Please go through reply 11 of anonym's referee-2.*

**Action:** *We are doing necessary corrections in language. We have also written a discussion section. Please go through reply 11 of anonymous refree-2.*

The responses to the specific comments are also presented as follows:

**Reply to the Specific comments**

**Comment 1:** This is a technical paper with a strong methodological focus. While the approaches used are well described, I would recommend the authors more clearly define the importance of the work in a broader hydrological context; for example, the relevance to hydrology and water resource management regionally and globally as this is only mentioned briefly.

**Reply 1:** *We have rewritten the introduction section and discussed about the time series analysis in a broader hydrological context following your suggestions (please go through the reply 1 of referee #2). Thank you very much for your constructive comments.*

**Action:** *We have rewritten the introduction section following the reviewer's suggestions. Please go through reply 1 of anonyms refree#2.*

**Comment 2:** This study is applied to a region in Bangladesh, however, both the approach and the paper would benefit if the transferability of the methodology could be highlighted by the authors; i.e. in what environments and under what conditions would the methods described work well.

**Reply 2:** *We are very pleased after going through your comments. ARIMA models (Box and Jenkins, 1976) are very much useful for forecasting hydrological variables such as rainfall (e.g., rainfall of the USA by Burlando et al., 1993), temperature (e.g., temperature of Bangladesh, Nury et al., 2017), PET (e.g., PET of Iran, Valipour, 2012), groundwater level (e.g., groundwater level of Canada by Adamowski and Chan), runoff (e.g. runoff of Russia by Nigam et al. 2014), water quality (water quality of Turkey by Faruk 2010) etc. These are the few examples of the application of ARIMA models in hydrology. However, ARIMA models have a limitation; these models cannot appropriately handle non-stationary hydrological data. Wavelet analysis is a suitable technique to overcome this*

*problem. Several studies have already demonstrated the advantages of wavelet analysis (Sang, 2013). Wavelet denoising has not attracted much attention in hydrologic science, though it has been used in the other science and engineering fields (Sang, 2013). We have discussed the advantages of denoising for forecasting the hydrological data in our article. Water balance components are related with a rage of hydro-meteorological variables. As we have given here few examples of worldwide applications of ARIMA models for forecasting the hydrological variables. Therefore, we may assume that our developed wavelet denoised ARIMA models can be applied for forecasting the hydrological variables worldwide. We will briefly discuss the matter in our new discussion section. Thank you very much for reminding us to write something about this important issue.*

**Action:** *As mentioned earlier, we have added a discussion section following the reviewer's suggestions. Please go through reply 11 of anonyms refree#2.*

**Comment 3:** The paper would also benefit from a separate limitations section; for example, as an additional section 3.4. Limitations, in addition to those found within the methods, should also be highlighted. This may include, for example, scarcity of data. Also, data from 11 stations are used to represent over 61,000 km2. A comment on how representative this data is would be welcome.

**Reply 3:** *As mentioned earlier, we will incorporate a discussion section where we highlight the limitations of our present study. We are doing another research work, and we hope that we will mention the scarcity of data and representativeness of the data for a large area in our next study.*

**Action:** *As mentioned earlier, we have added a discussion section. Please go through reply 11 of anonyms refree#2.*

**Comment 4:** Some of the text in Figure 6 is difficult to read. Also, please be consistent with the font.

**Reply 4:** *Thank you very much again for your valuable suggestions. We have already prepared new figures following your suggestions. We will add this figure in our final manuscript.*

**Action:** *We have prepared a new figure.*

[Figure]

**Figure 6: Plot of best wavelet ARIMA model first panel represents actual versus fitted values for the period of 1981-82 to 2012-2013, the second panel is normal Q-Q plot of residuals of the model, and the third panel shows actual, fitted and forecasted values for 2009-2010 to 2012-13 (a) $P_{ET}$ of Rangpur station located in north; (b) $A_{ET}$ of Ishurdi station located in the central part, (c) deficit of Rajshahi station located in NW Bangladesh and (d) surplus of Bhola station located in south of the study area.**

**Comment 5:** Some examples of typos and where sentence rephrasing would be beneficial:……

**Reply 5:** *We are going through the manuscript carefully and doing necessary corrections to improve the language. Thank you very much for your time and comments that help us a lot to improve the article.*

**Action**: *As mentioned earlier, we are doing necessary language corrections with the help of two professors of English department.*

**References**

Adamowski, J. and Chan, H. F. A wavelet neural network conjunction model for groundwater level forecasting. Journal of Hydrology, 407(1), 28–40, 2011.

Burlando, P., Rosso, R., Cadavid, L.G. Salas J. D.1993. Forecasting of short-term rainfall using ARMA models, Journal of Hydrology, Volume 144, Issues 1–4, April 1993, Pages 193-211. https://doi.org/10.1016/0022-1694(93)90172-6.

Faruk, D. Ö. 2010. A hybrid neural network and ARIMA model for water quality time series prediction, Engineering Applications of Artificial Intelligence 23: 586–594.

Kanoua, W. and Merkel B. J. (2015). Groundwater recharge in Titas Upazila in Bangladesh, Arab J Geosci 8:1361–1371, doi. 10.1007/s12517-014-1305-2.

Karim, M.R., Ishikawa, M. Ikeda, M. (2012) Modeling of seasonal water balance for crop production in Bangladesh with implications for future projection. Italian Journal of Agronomy 7(2). doi:10.4081/ija. 2012.e21.

McCabe, G.J., Markstrom, S.L. (2007) A monthly water-balance model driven by a graphical user interface: U.S. Geological Survey Open-File Report 2007–1088.

Nigam, R., Nigam, S., Mittal, S. K. 2014. The river runoff forecast based on the modeling of time series. Russian Meteorology and Hydrology, 39: 750. https://doi.org/10.3103/S1068373914110053.

Nury, A. H., Hasan, K. and Alam, J. B. 2017. Comparative study of wavelet-ARIMA and wavelet-ANN models for temperature time series data in northeastern Bangladesh, Journal of King Saud University – Science, 29, 47–61.

Valipour, M. 2012. Ability of Box-Jenkins Models to Estimate of Reference Potential Evapotranspiration (A Case Study: Mehrabad Synoptic Station, Tehran, Iran), IOSR Journal of Agriculture and Veterinary Science, ISSN: 2319-2380, ISBN: 2319-2372. Volume 1:5, PP 01-11.

Wolock, D. M. and McCabe, G. J. 1999. Effects of potential climatic change on annual runoff in the conterminous United States, Journal of the American Water Resources Association, 35, 1341–1350.

---

## Author Comment (AC2) · 2 Jan 2018

We are very much grateful to you for your valuable comments on our study "Modeling the Changes in Water Balance Components of Highly Irrigated Western Part of Bangladesh".

Overall comments: This manuscript describes the application of discrete wavelet transformation (DWT) and different forms of Mann-Kendal test to study changes in water balance components (WBCs). The authors also develop a "wavelet autoregressive moving average (ARIMA) model" to forecast WBCs. The contribution of the manuscript seems to be detecting trends and identifying periodicities in WBCs along with fore-

casting them after removing the noise from the time series. The manuscript is statistical than hydrological and I would say that hydrological concepts are insufficiently addressed and not fully developed. Moreover, there are some theoretical inaccuracies and confusing statements (especially in hydrologic side) that undermine the quality of this manuscript. Overall, the authors address an interesting subject; but in the current form, there are concerns and shortcomings that warrant major revisions.

Replay to overall comments: You are concerned about the hydrological theory. We think there are no inaccuracies in hydrological theory in our manuscript. We have explained the matter in the theoretical issues section; please go through the replies to this section.

Theoretical issues: Comment 1: There is confusion in the paper about the concept of "Water Balance" and its "Components". Water Balance Components (WBC) and some other parameters are frequently used in awkward and confusing sentences. As an example, potential evapotranspiration (PET), which is named as one of the WBCs in line 91 is called "the key parameter to estimate water balance components ..." in line 119. I strongly recommend that the authors provide the Water Balance Equation, briefly introduce Water Balance Components, and define which components they consider in their study, clearly. They may explain these concepts at the beginning of the "Methods" section (section 2.3). They may also mention the reason(s) for selecting each WBC. Comment 2: Following the previous comment, both Potential Evapotranspiration (PET) and Actual Evapotranspiration (AET) are considered, surprisingly, as components of water balance equation (for example in lines 91, 265, 325) without any explanation on their application and role in the equation. However, the application of these parameters in Water Balance Equation is different and they cannot be considered both at the same time. I would also suggest the authors revise their manuscript to ensure that no confusing sentence remains on this subject.

Reply to comments 1 and 2: You are concerned about the water balance components (WBCs)and the input parameters of WBCs. We think we have appropriately presented

the WBCs and the input parameters of WBCs in our manuscript. There are a lot of articles on water balance/ WBCs, however, please go through the following few sentences for your clarifications about the water balance components and input parameters of water balance components. McCabe and Wolock (2013) studied on "Temporal and spatial variability of the global water balance". Please go through the first sentence of the abstract. You will find that precipitation, actual evapotranspiration (AET), runoff, and potential evapotranspiration (PET) are water balance components. Therefore, we hope that you will understand PET is one of the WBCs. It is also one of the important input parameters of WBCs to calculate other WBCs like AET, surplus/ runoff and so on. For your clarification, please go through the study conducted by Xu and Singh (1998). This is a review article on water balance models. We have given here only two examples for your clarification about the hydrological theory. We hope that you will find a lot of articles on water balance/ WBCs and there are some citations on water balance study in our manuscript too. We will add water balance component equations in our final manuscript. However, we will not add the PET equation as it is a well-established method. Moreover, PET has been calculated by Penman-Monteith (Allen et al., 1998) method and there is a citation in the manuscript.

Comment 3: In line 145, the authors stated that "When rainfall is greater than PET the soil always remains full of water and:", which is an inaccurate statement. I understand the authors try to explain the concept of surplus; however, surplus occurs when the soil becomes saturated and infiltration is hardly possible.

Reply 3: Thank you very much to find out this mistake. We will incorporate this correction in our final manuscript.

Title and Abstract Comment 4: The authors should perhaps reframe the title to better reflect their work. The present title implies that the study is mainly concentrated on the interaction between changes in water balance components and intensive irrigation in Western Bangladesh.

Reply 4: We will not reframe the title. WBCs such as PET, AET, Surplus and deficit are related to the crop water requirements, irrigation requirement, irrigation scheduling and so on. Therefore, we will not reframe our title. Comment 5: I would recommend that the authors name water balance components that they consider in this study in the abstract. They may provide then a summary of the methodology and results in a more organized way.

Reply 5: We have mentioned the names of the WBCs which are considered in our study. Please go through the abstract, you will find these. We do not think it is necessary to add descriptions of water balance calculation process in the abstract section. Moreover, there is a methodology section.

Comment 6: I was wondering whether the authors apply ARIMA or ARMA models in their study. In case of having ARIMA, which stands for "Autoregressive Integrated Moving Average" they should revise the statement in line 17.

Reply 6: Thank you very much to find out this mistake. We have used ARIMA model. We will incorporate this correction to our final manuscript.

Comment 7: Line 34: The statement ": : : findings of study can be used to improve water resources management : : :" is too generic. Please clarify in what respect this study can improve water resources management in the highly irrigated area.

Reply 7: There is a lot of information on WBCs, trends and periodicity in WBCs and a new developed methodology for the forecasting WBCs. We hope that water resources manager will get a lot of information from our study. Please also go to the reply 4 for your understanding how WBCs are related to the water management.

The Structure: Comment 8: In general, the paper has no flow and each section seems to be a separate part without proper connection to the other sections. I think the authors should improve the structure and flow of their manuscript.

Reply 8: Thank you very much for your suggestions. Our final manuscript will be a

better organized one.

Comment 9: I believe that the "Introduction" should be significantly revised. For instance, the literature review on periodicity and using wavelet transformation is only limited to few sentences. The authors can elaborate more on what the previous researchers have done and how this study differs from previous attempts.

Reply 9: We think we have to emphasize on time series analysis instead of the periodicity only. Our main objective was to develop a methodology for forecasting the WBCs. Moreover, there are some citations related to the topics in introduction and methodology sections.

Comment 10: In section 2.2, "Data", I would suggest that authors provide the time duration they used in this study.

Reply 10: Thank you very much for the suggestion. Though there is no information on time duration in the data section, we have mentioned it in the first sentence of results of analysis section. We will also add time duration in the data section.

Comment 11: Headings are awkward and in some cases poorly selected. For example, in lines 265 and 325, (sections 3.2.1, and 3.2.2) it would be better to replace "PET" with "Potential Evapotranspiration" and "AET" with "Actual Evapotranspiration" respectively.

Reply 11: We will replace these headings in the final manuscript.

Comment 12: In section 2.3, "Methods", I would suggest that the authors provide a general overview of their methods and then explain each section in detail rather than starting the section immediately with a sub-heading.

Reply 12: We will add a general overview of the methodology. Please go through the reply 7 of referee #2.

Comment 13: Section 3, which seems to provide results of the study, is poorly structured. Sentences are awkward and poorly written, which makes it difficult for readers

to follow.

Reply 13: We will do the necessary language corrections.

Comment 14: I was wondering why the authors consider the section "Model Selection and Forecasting Ability" as a sub-heading of Results section (line 352). The methodology of the modeling and considerations regarding model selection should be discussed in the "Methods" section.

Reply 14: We think the heading of this sub-section is right. Please also go through reply 15.

Comment 15: Following the above-mentioned comment, section 3.3 (lines 352-416) contains the model selection, methodology, results, and some discussions. The section is too long and without proper flow. I suggest the authors break this section into methodology, results, and discussion to help readers better follow their work.

Reply 15: Thank you very much for your valuable suggestion. Though there are some descriptions linked to methodology, we think we do not need to rewrite this section as our manuscript is methodological in nature. Therefore, it is necessary to link this section with the methodology.

Comment 16: The authors use passive voice and active sentences alternatively in the manuscript. They may re-write these complicated parts. For example lines 293-294.

Reply 16: As we have mentioned earlier, we will do the necessary language corrections.

Comment 17: In the "Summary and Conclusion", the authors mostly repeat some parts of the manuscript. I would expect to read a more conclusive summary and conclusion. For example, in Lines 447-449, (as mentioned earlier in comments on the Abstract) the authors stated that results of this study "can be incorporated to water resources management plans : : :"; but they didn't explain how this incorporation would take place. I suggest that authors add some explanations to the manuscript to clarify in

what respect their work will affect water resources management in the highly irrigated lands.

Reply 17: We will make the necessary corrections, please go also through the reply 7. Moreover, it is not only conclusion section. Please look at the heading of this section, it is Summary and Conclusion.

Comment 18: Lines 43-44: confusing and awkward statement: "Two important climatic variables like rainfall and PET that derives from the climatic variables are the main inputs in the water balance modeling". Please re-write the sentence.

Reply 18: You may understand this is a technically correct sentence after going through our replies. Comment 19: Lines 74-77: Please re-write the statement.

Reply 19: We will check these sentences before final submission. Comment 20: Lines 81-82: ": : : most of studies were limited to detect trends or forecasting of rainfall and temperature and few studies on PET and water balance." References are required.

Reply 20: At first, we have discussed about the relevant studies in Bangladesh. Therefore, it is not a separate sentence. There are some references in the manuscript. Please go through the manuscript carefully.

Comment 21: In section 2.1, it is stated that rice, the main crop cultivated in Bangladesh is mainly rain-fed or irrigated by groundwater resources (lines 104-106). Unfortunately, the authors have not clearly explained the relation between their study and irrigated area or even irrigation water demand in the study area. They may define how their work will affect the "Highly Irrigated Western Part of Bangladesh".

Reply 21: Please go to reply 7.

Comment 22: Lines 144-147, as acknowledged earlier, the statement needs theoretical revision. However, references are required for the definitions of surplus and deficit.

Reply 22: We will check these sentences.
Reply 23: For the statement in lines 147-151, on the AET and its "calculation", references are required.

Reply 23: There is a reference in line 137.

Comment 24: Lines 398-400, awkward sentence. Please re-write this sentence.

Comment 25: In general, the writing can be significantly improved. . ...

Reply to 24 & 25: As we have mentioned earlier, we will do the necessary language corrections. Thank you very much for your suggestions.

References Allen, R. G., Pereira, L. S., Raes, D. and Smith, M. Crop evapotranspiration: guidelines for computing crop water requirements. FAO Irrigation and Drainage Paper, No. 56, Rome, Italy, p.328, 1998.

McCabe, G.J. and Wolock, D.M. Temporal and spatial variability of the global water balance, 120, 375, Climatic Change, https://doi.org/10.1007/s10584-013-0798-0, 2013.

Xu, C.Y. and Singh, V.P. A Review on Monthly Water Balance Models for Water Resources Investigations, Water Resources Management 12, 20, https://doi.org/10.1023/A:1007916816469, 1998.

Please also note the supplement to this comment:
https://www.hydrol-earth-syst-sci-discuss.net/hess-2017-523/hess-2017-523-AC2-supplement.pdf

**Supplement:**

Anonyms Referee # 2

Responses to interactive comments of anonymous referee (#2) about our study "**Modeling the Changes in Water Balance Components of Highly Irrigated Western Part of Bangladesh**"

**General Comments:**

This paper investigates changes in water balance components between 1981-82 to 2012-13 in an area of Bangladesh that is intensively irrigated. First, historical trends are examined using the Mann-Kendal test and discrete wavelet transformation. Then, ARIMA models are developed in order to forecast changes in water balance components. The paper produces some interesting results; particularly around the use of ARIMA models that are fitted to wavelet denoised time series data.

The paper is well organised and about the right length for a study of this kind. Generally speaking the equations are well laid out and easy to follow. However, as it stands the level of English used in the paper is poor, which makes some sections very difficult to follow. I would strongly advise the authors to consult a proofreader who has full professional proficiency in written English. Nevertheless in my judgement the scientific content is sound and represents an interesting approach to analysing and forecasting changes in the water balance. Thus, I would reconsider this paper for publication following a major revision to improve the quality of English as well as addressing the specific points mentioned below.

**Reply to general comments:**

*We are very much grateful to you for your valuable comments about our study '**Modeling the Changes in Water Balance Components of Highly Irrigated Western Part of Bangladesh**'. We have already gone through your comments and we will incorporate the necessary corrections in the relevant sections. We are also doing necessary corrections in language which is your main concern regarding the manuscript. Actually, we will receive help from two professors of English for doing the corrections in our manuscript. Thank you very much for your suggestions that will help us to prepare a well-organized paper. Yes, the paper is methodological in nature; we have tried to forecast water balance components (WBCs) more preciously after denoising the time series by discrete wavelet transformation. We also expect that there is a scope for further improvement of the methodology by different combinations of wavelet techniques. We will add a discussion section following your suggestions.*

**Action:** *We have written a discussion section (3.4). Please go to reply 11.*

The responses to the specific comments are also presented as follows:

**Reply to the Specific comments**

**Comment 1:** The Introduction lacks focus, does not provide much critical analysis and does not place the work in the broader context of water resources management. I would like to see the aim of the paper clearly stated in the first paragraph of the paper, so that readers know what the paper is setting out to achieve. The paper is methodological in nature, so the Introduction must make clear to the reader the state-of-the-art in time series analysis for water resources management. Line 82 onwards does include some critical analysis but, in my opinion, it is insufficient to persuade the reader of the approach and its relevance to hydrology more generally. Focus less on the results of studies and instead examine and compare the different ways that previous researchers have tackled the problem.

**Reply 1:** *We also agree with your comments that we need to give more emphasis on the state-of-the-art in time series analysis for water resources management in the introduction section. Thus, we will revise the introduction section following your suggestions.*

**Action:** *We have rewritten the introduction section (1) following the reviewers suggestions.*

**1. Introduction**

[revised manuscript text omitted]

**Comments 2 to 6:**

**Comment 2:** The first half of Section 2.3.1 is probably superfluous: it is well known that PenmanMonteith is the most appropriate method to use to calculate PET, data permitting.

**Comment 3:** Line 137: I'm not exactly sure what 'Deficit' and 'Surplus' mean in this context (nor is it clear why they are capitalised) – provide additional explanation.

**Comment 4:** Line 137-139: It is presented as a fact that 'the concept of water balance in the unsaturated zone...give the best estimation for the real world' - this is quite a statement and surely unjustified. I note that Bakundukize et al 2011 were investigating hydrology in Burundi – are there similarities to Bangladesh? Provide some additional arguments for using the Thornthwaite and Mather model.

**Comment 5:** Line 143: Wolock and McCabe (1999) examined hydrology in the United States – is it reasonable to assume a 5% runoff in Bangladesh, given its tropical climate?

**Comment 6:** Line 145-151: Express the water balance model as equations. The calculation of the water balance is fundamental to the subsequent analysis, so it should be clear what you have done.

**Reply to comments 2-6:**

*Thank you very much for your valuable comments. These comments are related to the section 'Calculation of $P_{ET}$ and WBC (2.3.1)'. We will rewrite this section following your suggestions. Line 137.-we will also add a brief*

*description of $A_{ET}$, deficit and surplus of water. In line 143, firstly, direct runoff (DRO) is not the total runoff. It is the fraction of rainfall that immediately enters low-lying areas and/or stream channels because of infiltration-excess flow is known as DRO. "The fraction of $P_{rain}$ that becomes DRO is specified; based on previous water-balance analyses, 5 percent is a typical value to use (Wolock and McCabe, 1999)". This concept has also been applied to estimate the direct runoff in Bangladesh and yields good results (Karim et al., 2012; Kanoua and Merkel, 2015). About line 145-151, we also agree with you and grateful to you for your critical findings. We will add the equations of water balance components in the main manuscript. However, we may not add the Penman-Monteith equation (Allen et al., 1998) as it is a well-established method.*

**Action:** *We have rewritten the section 2.3.1 following the reviewer's suggestions.*

**2.3.1 Calculation of $P_{ET}$ and WBCs**

*Potential evapotranspiration is the key parameter to estimate WBCs. It has been calculated by Penman-Monteith equation (Allen et al., 1998) in the present study. The soil-water balance concept proposed by Thornthwaite and Mather (1955) is one of the most widely used methods for estimating the WBCs. It is suitable for assessing the effectiveness of agricultural water resources management practices and regional water balance studies as it allows estimating the actual evapotranspiration ($A_{ET}$), water deficit and surplus (e.g., Chapman and Brown 1966, Bakundukize et al., 2011, Karim et al., 2012, Viaroli et al., 2017). $A_{ET}$ is the amount of water which is removed from the surface due to the process of evaporation and transpiration. The amount by which $P_{ET}$ exceeds $A_{ET}$ is termed as deficit and surplus is the excess rainfall after the soil has reached its water holding capacity (de Jong and Bootsma, 1997). It is necessary to calculate the field capacity of the soil for estimating the WBCs. Field capacity of soil in the study area has been calculated using the soil texture map of Bangladesh prepared by Soil Resource Development Institute of Bangladesh (SRDI, 1998) and the description of soils of Bangladesh presented by Huq and Shoaib (2013). Thornthwaite and Mather (1957) suggested values for water holding capacity of soil and rooting depth of the plants have been used for WBCs estimation in the present study. The first step of the calculation is the subtraction of 5% rainfall from the monthly rainfall data as this amount of water has been lost due to direct runoff (Wolock and McCabe, 1999; Karim et al., 2012; Kanoua and Merkel, 2015). The remaining amount of rainfall has been included in the calculation. The WBCs like $A_{ET}$, surplus and deficit have been estimated based on the following formulas presented in Table 1:*

**Table 1**: Calculations of water balance components (Thornthwaite and Mather, 1957)

| | | Wet Season | | Dry Season |
|---|---|---|---|---|
| | | $Surplus = (P - R_0) - P_{ET} > 0$ | | $Surplus = (P - R_0) - P_{ET} < 0$ |
| | $S_B = C_{AP}$ | $S_B < C_{AP}$ | | |
| | | $(P - R_0) - P_{ET} \leq C_{AP} - S_B$ | $(P - R_0) - P_{ET} > C_{AP} - S_B$ | |
| $S_B$ | $C_{AP}$ | $S_B + (P - R_0) - P_{ET}$ | $C_{AP}$ | $C_{AP} * e^{-APWL/C_{AP}}$ |
| $A_{ET}$ | $P_{ET}$ | $P_{ET}$ | $P_{ET}$ | $(P - R_0) + \Delta S_B$ |
| Deficit | 0 | 0 | 0 | $P_{ET} - A_{ET}$ |

*Here, P= Rainfall, $R_0$= Direct runoff, $P_{ET}$=Potential evapotranspiration, $A_{ET}$=Actual evapotranspiration, APWL=Accumulated potential water loss, $S_B$=Water stored in soil: $S_B=C_{AP}*e^{-APWL/CAP}$, $C_{AP}$= Soil capacity: average rooting depth* water content at field capacity and $\Delta S_B$ = Changes in $S_B$*

**Comment 7.** In my view Section 2 would benefit from a short overview describing the reason for carrying out the various steps (water balance, Mann-Kendal, wavelet analysis, ARIMA) and how they relate to one another. At the moment this is not clear.

**Reply 7:** *We will incorporate a short overview in the section 2.3. However, there are descriptions on water balance, wavelet analysis and ARIMA model in sections 2.3.1, 2.3.3 and 2.3.4 respectively. Therefore, we will only revise the trend test section (2.3.2).*

**Action:** *We have written a short over view of the methods in section 2.3 and rewritten the section 2.3.2 following the reviewer's suggestions.*

*2.3 Methods*

*In the present study, WBCs have been calculated and trends in WBCs have been identified by MK/MMK test for evaluating the long-term water balance of the highly irrigated western part of Bangladesh. DWT data of WBCs time series have been analyzed for identifying the time period responsible for the trend in the data. WBCs have been forecasted by ARIMA models and the model performance has been evaluated statistically. If the performance of the model is not satisfactory for forecasting the WBCs, original time series has been denoised using discrete wavelet transformation techniques to improve the performance of the model. The descriptions of the methods have been presented in the following sections:*

*2.3.2 Trend Test*

*In the present study, the trends in WBCs have been detected by non-parametric Mann–Kendall (MK) (Mann, 1945; Kendal, 1975) test as it shows better performance to identify trends in hydrological variables like rainfall (e.g. Shahid, 2010), temperature (e.g. Kamruzzaman et al., 2016a), $P_{ET}$ (e.g. Kumar et al., 2016), soil moisture (e.g. Tabari and Talaee, 2013), runoff (e.g. Pathak et al., 2016), groundwater level (e.g. Rahman et al., 2016), water quality (e.g. Lutz et al., 2016) in comparison to the parametric test (Nalley et al., 2012). MK test cannot appropriately calculate the test statistic (Z) due to underestimating the variance (Hamed and Rao, 1998) if there is a significant serial correlation at lag-1 in the time series data (Yue et al., 2002). The lag-1 auto-correlation has been checked before analyzing the time series data if there is a significant lag-1 auto-correlation at 5% level, the Modified MK (Hamed and Rao, 1998) has been applied instead of MK test. The estimated Z statistic of MK/MMK test has been evaluated for the direction of the trend such as positive Z statistic to indicate increasing trend and vice versa. Moreover, it also indicates the level of significance of the obtained trend, for example, if the calculated Z statistic is equal to or greater than the tabulated value of Z statistic +1.96 that indicates a significant positive trend at 95% confidence level or if it is equal to or less than -1.96 that indicates a significant decreasing trend. Moreover, the sequential values of u(t) statistic of MK test derived from the progressive analysis of MK test (Sneyers, 1990), u(t) is similar to the Z statistic (Partal and Küçük, 2006), have been used for investigating the change point detection. The magnitude of the change has been calculated by Sen's slope estimator (Sen, 1968). There are many good explanations (notably Nalley et al., 2012) of these methods mentioned in this section and details regarding these, furthermore, can be referred to Mann (1945); Sen (1968); Kendall (1971); Hamed and Rao (1998); Sneyers (1990); Yue et al. (2002).*

**Comment 8.** Line 154: Which hydrological variables were investigated?

**Reply 8:** *Hydrological variables like rainfall, temperature, $P_{ET}$, runoff, groundwater level and water quality have been investigated by MK test to detect trends in time series data. We have mentioned about these in reply 7 (revised section 2.3.2).*

**Action:** *As we mentioned in reply 7, we have rewritten the section 2.3.2.*

**Comment 9.** Line 159: What are these 'Z' values, and what is their importance? This is the first the time they have been mentioned.

**Reply 9:** *Thank you very much for noticing the Z statistic. We have incorporated text about Z statistic in reply 7 (revised section 2.3.2).*

**Action:** *As mentioned earlier, we have rewritten the section 2.3.2 and added necessary text on Z statistic.*

**Comment 10.** Section 2.3.7 is unnecessary here unless it specifically influences the scientific results. Instead put this information in an Appendix or similar (however, I congratulate the authors for putting their computer code alongside the paper – this is not done often enough).

**Reply 10:** *This section will be moved to electronical supplementary material (ESM).*

**Action:** *This section will be moved to electronical supplementary material.*

**Comment 11:** I think Section 3 should simply describe the results, with an additional 'Discussion' section for placing the results in the context of other studies (e.g. Line 281, 288, 321 etc. should be put in a Discussion section). The discussion should include additional analysis discussing the various limitations and weaknesses of the present study as well as suggesting improvements.

**Reply 11:** *We are grateful to you for your valuable comments. We will incorporate a discussion section (3.4) after the results of analysis following your suggestions. We also hope that this section will help readers about the results described in the manuscript and how can we improve the performance of the model.*

**Action:** *We have written a discussion section (3.4) following the suggestions of the reviewers.*

**3.4 Discussion**

[revised manuscript text omitted]

**Comment 12**: Line 265: Use 'Potential evapotranspiration' rather than 'Pet' in section headings.

**Reply 12:** *We will incorporate your suggestion. We will replace $P_{ET}$ by Potential Evapotranspiration (3.2.1) and $A_{ET}$ by Actual Evapotranspiration (3.2.2).*

**Action:** *We will replace $P_{ET}$ by Potential Evapotranspiration (3.2.1) and $A_{ET}$ by Actual Evapotranspiration (3.2.2) in the heading.*

**Comment 13.** Line 359: This is just a piece of computer code – what does it do, and what insight does it provide that you cannot gain from manual interpretation of ACF, PACF, AIC, BIC?

**Reply to Comment 13:** *ACF, PACF, AIC, BIC are important parameters for selection of an accurate ARIMA model for forecasting. For manual model sections, we need to find out the best combinations of these parameters with acceptable error. Besides manual model selections, automatic model selection option of the forecast package of R (R-language software) has been used in the present study. This option helps us find out the best model, especially when we could not find a satisfactory model (model with acceptable error) by manual interpretation of ACF, PACF, AIC and BIC.*

**Action:** *We have added the answer here for the reviewer.*

**Comment 14.** Line 362: What is a Q-Q plot?

**Reply 14:** *The quantile-quantile (Q-Q) plot is a probability plot to check the hypothesis of normality for a certain samples. It is graphical method which compare between two probability distributions based on the quantile values (Filliben, 1975). In our study, we have prepared Q-Q plot to check the normality of residuals.*

**Action:** *We have added the answer here for the reviewer.*

**Comment 15.** Line 386-416: To my mind this passage is the strongest part of the paper -the discussion should emphasise this result and its relevance to water resources management more generally.

**Reply to Comment 15:** *Thank you very much again for your observations and comments. We will add a discussion section as we have mentioned and added in reply 11.*

**Action:** *As mentioned earlier, we have added a discussion section (3.4). Please go to the reply 11.*

**Comment 16:** As I mentioned earlier, I would strongly suggest creating an additional Discussion section in which to discuss the results in the context of other studies, highlight limitations and propose future research directions.

**Reply to Comment 16:** *We have mentioned the matter earlier. We are grateful to you for your comments that help us improve the quality of our present research work. Thank you very much again.*

**Action:** *As mentioned earlier, we have added a discussion section (3.4). Please go to the reply 11.*

[revised manuscript text omitted]

---

## Author Comment (AC3) · 3 Jan 2018

We are very much grateful to you for your valuable comments about our study 'Modeling the Changes in Water Balance Components of Highly Irrigated Western Part of Bangladesh'. Thank you very much for your suggestions that will help us to prepare the final manuscript. Detailed responses to the general and specific comments are attached as a supplement.

---

## Author Response (AR1)

**Anonyms Referee # 1**

We thank the referee (#1) for reviewing our manuscript entitled 'Modeling the Changes in Water Balance
Components of Highly Irrigated Western Part of Bangladesh' and for his/ her valuable comments to improve
our manuscript. We have responded to referee (#1) comments below:

**Overall Comments:**
This paper by A.T.M. Sakiur Rahman et al (the authors) describes the development of wavelet autoregressive
moving average models to forecast changes in water balance components. The approach is applied to a highly-
irrigated area in Western Bangladesh using data collected between 1982 and 2013. The authors show that the
approach can be used to forecast short term changes in water balance components but suggest that models could
be further improved using different combinations of wavelet analysis.
**Reply to overall comments:**
*We are very much grateful to you for your valuable comments about our study* **'Modeling the Changes in Water**
**Balance Components of Highly Irrigated Western Part of Bangladesh***'. We have gone through your comments*
*and we will incorporate the necessary corrections in the relevant sections. We are also doing necessary*
*corrections in language which is your main concern regarding the manuscript. Thank you very much for your*
*suggestions that will help us to prepare a good paper. Yes, the paper is methodological in nature; we have tried*
*to forecast water balance components (WBCs) more preciously after denoising the time series by discrete*
*wavelet transformation. We also expect that there is a scope for further improvement of the methodology by*
*different combinations of wavelet techniques. We will add a discussion section (3.4) where we discuss the*
*advantages and limitations of our study. Please go through reply 11 of anonym's referee-2.*
**Action:** *We are doing necessary corrections in language. We have also written a discussion section. Please go*
*through reply 11 of anonymous refree-2.*
The responses to the specific comments are also presented as follows:
**Reply to the Specific comments**
**Comment 1:** This is a technical paper with a strong methodological focus. While the approaches used are well
described, I would recommend the authors more clearly define the importance of the work in a broader
hydrological context; for example, the relevance to hydrology and water resource management regionally and
globally as this is only mentioned briefly.
**Reply 1:** *We have rewritten the introduction section and discussed about the time series analysis in a broader*
*hydrological context following your suggestions (please go through the reply 1 of referee #2). Thank you very*
*much for your constructive comments.*
**Action:** *We have rewritten the introduction section following the reviewer's suggestions. Please go through*
*reply 1 of anonyms refree#2.*
**Comment 2:** This study is applied to a region in Bangladesh, however, both the approach and the paper would
benefit if the transferability of the methodology could be highlighted by the authors; i.e. in what environments
and under what conditions would the methods described work well.
**Reply 2:** *We are very pleased after going through your comments. ARIMA models (Box and Jenkins, 1976) are*
*very much useful for forecasting hydrological variables such as rainfall (e.g., rainfall of the USA by Burlando et*
*al., 1993), temperature (e.g., temperature of Bangladesh, Nury et al., 2017), PET (e.g., PET of Iran, Valipour,*
*2012), groundwater level (e.g., groundwater level of Canada by Adamowski and Chan), runoff (e.g. runoff of*
*Russia by Nigam et al. 2014), water quality (water quality of Turkey by Faruk 2010) etc. These are the few*
*examples of the application of ARIMA models in hydrology. However, ARIMA models have a limitation; these*
*models cannot appropriately handle non-stationary hydrological data. Wavelet analysis is a suitable technique*
*to overcome this problem. Several studies have already demonstrated the advantages of wavelet analysis (Sang,*
*2013). Wavelet denoising has not attracted much attention in hydrologic science, though it has been used in the*
*other science and engineering fields (Sang, 2013). We have discussed the advantages of denoising for*
*forecasting the hydrological data in our article. Water balance components are related with a rage of hydro-*

*meteorological variables. As we have given here few examples of worldwide applications of ARIMA models for forecasting the hydrological variables. Therefore, we may assume that our developed wavelet denoised ARIMA models can be applied for forecasting the hydrological variables worldwide. We will briefly discuss the matter in our new discussion section. Thank you very much for reminding us to write something about this important issue.*

**Action:** *As mentioned earlier, we have added a discussion section following the reviewer's suggestions. Please go through reply 11 of anonyms refree#2.*

**Comment 3:** The paper would also benefit from a separate limitations section; for example, as an additional section 3.4. Limitations, in addition to those found within the methods, should also be highlighted. This may include, for example, scarcity of data. Also, data from 11 stations are used to represent over 61,000 km2. A comment on how representative this data is would be welcome.

**Reply 3:** *As mentioned earlier, we will incorporate a discussion section where we highlight the limitations of our present study. We are doing another research work, and we hope that we will mention the scarcity of data and representativeness of the data for a large area in our next study.*

**Action:** *As mentioned earlier, we have added a discussion section. Please go through reply 11 of anonyms refree#2.*

**Comment 4:** Some of the text in Figure 6 is difficult to read. Also, please be consistent with the font.

**Reply 4:** *Thank you very much again for your valuable suggestions. We have already prepared new figures following your suggestions. We will add this figure in our final manuscript.*

**Action:** *We have prepared a new figure.*

[Figure]

**Figure 6: Plot of best WD-ARIMA model first panel represents actual versus fitted values for the period of 1981-82 to 2012-2013, the second panel is normal Q-Q plot of residuals of the model, and the third panel shows actual, fitted and forecasted values for 2009-2010 to 2012-13 (a) $P_{ET}$ of Rangpur station located in north; (b) $A_{ET}$ of Ishurdi station located in the central part, (c) deficit of Rajshahi station located in NW Bangladesh and (d) surplus of Bhola station located in south of the study area.**

**Comment 5:** Some examples of typos and where sentence rephrasing would be beneficial:……

**Reply 5:** *We are going through the manuscript carefully and doing necessary corrections to improve the language. Thank you very much for your time and comments that help us a lot to improve the article.*

**Action**: *As mentioned earlier, we are doing necessary language corrections with the help of two professors of English department.*

Bangladesh that is intensively irrigated. First, historical trends are examined using the Mann-Kendal test and
discrete wavelet transformation. Then, ARIMA models are developed in order to forecast changes in water
balance components. The paper produces some interesting results; particularly around the use of ARIMA
models that are fitted to wavelet denoised time series data.

The paper is well organised and about the right length for a study of this kind. Generally speaking the equations
are well laid out and easy to follow. However, as it stands the level of English used in the paper is poor, which
makes some sections very difficult to follow. I would strongly advise the authors to consult a proofreader who
has full professional proficiency in written English. Nevertheless in my judgement the scientific content is
sound and represents an interesting approach to analysing and forecasting changes in the water balance. Thus, I
would   reconsider this paper for publication following a major revision to improve the quality of English as
well as addressing the specific points mentioned below.

**Reply to general comments:**

*We are very much grateful to you for your valuable comments about our study* **'Modeling the Changes in Water**
**Balance Components of Highly Irrigated Western Part of Bangladesh***'. We have already gone through your*
*comments and we will incorporate the necessary corrections in the relevant sections. We are also doing*
*necessary corrections in language which is your main concern regarding the manuscript. Actually, we will*
*receive help from two professors of English for doing the corrections in our manuscript. Thank you very much*
*for your suggestions that will help us to prepare a well-organized paper. Yes, the paper is methodological in*
*nature; we have tried to forecast water balance components (WBCs) more preciously after denoising the time*
*series by discrete wavelet transformation. We also expect that there is a scope for further improvement of the*
*methodology by different combinations of wavelet techniques. We will add a discussion section following your*
*suggestions.*

**Action:** *We have written a discussion section (3.4). Please go to reply 11.*

The responses to the specific comments are also presented as follows:

**Reply to the Specific comments**

**Comment 1:** The Introduction lacks focus, does not provide much critical analysis and does not place the work
in the broader context of water resources management. I would like to see the aim of the paper clearly stated in
the first paragraph of the paper, so that readers know what the paper is setting out to achieve. The paper is
methodological
in nature, so the Introduction must make clear to the reader the state-of-the-art in time series analysis for water
resources management. Line 82 onwards does include some critical analysis but, in my opinion, it is insufficient
to persuade the reader of the approach and its relevance to hydrology more generally. Focus less on the results
of
studies and instead examine and compare the different ways that previous researchers have tackled the problem.

**Reply 1:** *We also agree with your comments that we need to give more emphasis on the state-of-the-art in time*
*series analysis for water resources management in the introduction section. Thus, we will revise the*
*introduction section following your suggestions.*

**Action:** *We have rewritten the introduction section (1) following the reviewers suggestions.*

*1. Introduction*

[revised manuscript text omitted]

**Comments 2 to 6:**

**Comment 2:** The first half of Section 2.3.1 is probably superfluous: it is well known that PenmanMonteith is the most appropriate method to use to calculate PET, data permitting.

**Comment 3:** Line 137: I'm not exactly sure what 'Deficit' and 'Surplus' mean in this context (nor is it clear why they are capitalised) – provide additional explanation.

**Comment 4:** Line 137-139: It is presented as a fact that 'the concept of water balance in the unsaturated zone...give the best estimation for the real world' - this is quite a statement and surely unjustified. I note that Bakundukize et al 2011 were investigating hydrology in Burundi – are there similarities to Bangladesh? Provide some additional arguments for using the Thornthwaite and Mather model.

**Comment 5:** Line 143: Wolock and McCabe (1999) examined hydrology in the United States – is it reasonable to assume a 5% runoff in Bangladesh, given its tropical climate?

**Comment 6:** Line 145-151: Express the water balance model as equations. The calculation of the water balance is fundamental to the subsequent analysis, so it should be clear what you have done.

**Reply to comments 2-6:**

*Thank you very much for your valuable comments. These comments are related to the section 'Calculation of $P_{ET}$ and WBC (2.3.1)'. We will rewrite this section following your suggestions. Line 137.-we will also add a brief description of $A_{ET}$, deficit and surplus of water. In line 143, firstly, direct runoff (DRO) is not the total runoff. It is the fraction of rainfall that immediately enters low-lying areas and/or stream channels because of infiltration-excess flow is known as DRO. "The fraction of $P_{rain}$ that becomes DRO is specified; based on previous water-balance analyses, 5 percent is a typical value to use (Wolock and McCabe, 1999)". This concept has also been applied to estimate the direct runoff in Bangladesh and yields good results (Karim et al., 2012; Kanoua and Merkel, 2015). About line 145-151, we also agree with you and grateful to you for your critical*

*findings. We will add the equations of water balance components in the main manuscript. However, we may not*
*add the Penman-Monteith equation (Allen et al., 1998) as it is a well-established method.*

**Action:** *We have rewritten the section 2.3.1 following the reviewer's suggestions.*

***2.3.1 Calculation of Potential evapotranspiration and Water Balance Components***

*Potential evapotranspiration ($P_{ET}$) is the key parameter to estimate WBCs. It has been calculated by Penman-*
*Monteith equation (Allen et al., 1998) in the present study. The soil-water balance concept proposed by*
*Thornthwaite and Mather (1955) is one of the most widely used methods for estimating the WBCs. It is suitable*
*for assessing the effectiveness of agricultural water resources management practices and regional water*
*balance studies as it allows estimating the actual evapotranspiration ($A_{ET}$), water deficit and surplus (e.g.,*
*Chapman and Brown 1966, Bakundukize et al., 2011, Karim et al., 2012, Viaroli et al., 2017). $A_{ET}$ is the amount*
*of water which is removed from the surface due to the process of evaporation and transpiration. The amount by*
*which $P_{ET}$ exceeds $A_{ET}$ is termed as deficit and surplus is the excess rainfall after the soil has reached its water*
*holding capacity (de Jong and Bootsma, 1997). It is necessary to calculate the field capacity of the soil for*
*estimating the WBCs. Field capacity of soil in the study area has been calculated using the soil texture map of*
*Bangladesh prepared by Soil Resource Development Institute Bangladesh (SRDI, 1998) where the description*
*of soils has been presented by Huq and Shoaib (2013). The values for water holding capacity of soil and rooting*
*depth of the plants suggested by Thornthwaite and Mather (1957) have been used for WBCs estimation in the*
*present study. The first step of the calculation is the subtraction of 5% rainfall from the monthly rainfall data as*
*this amount of water has been lost due to direct runoff (Wolock and McCabe, 1999; Karim et al., 2012; Kanoua*
*and Merkel, 2015). The remaining amount of rainfall has been included in the calculation. The WBCs like $A_{ET}$,*
*surplus and deficit have been estimated based on the formulas presented in Table 1 and details of WBCs*
*calculation can be found in Electronically Supplementary Martial (EMS).*

**Table 1**: Calculations of water balance components (Thornthwaite and Mather, 1957)

|  | Wet months $(P - R_0) > P_{ET}$ | Dry months $(P - R_0) < P_{ET}$ |
|---|---|---|
| $A_{ET}$ | $P_{ET}$ | $(P - R_0) + \Delta S_B$ |
| Deficit | 0 | $P_{ET} - A_{ET}$ |
| Surplus | $(P - R_0) - P_{ET}$ | 0 |

Where $P$ is the rainfall (mm), $R_0$ is the direct runoff (mm), $P_{ET}$ is the potential evapotranspiration (mm), $A_{ET}$
is the actual evapotranspiration (mm) and $\Delta S_B$ is the changes in soil moisture storage (mm).

**Comment 7.** In my view Section 2 would benefit from a short overview describing the reason for carrying out
the various steps (water balance, Mann-Kendal, wavelet analysis, ARIMA) and how they relate to one another.
At the moment this is not clear.

**Reply 7:** *We will incorporate a short overview in the section 2.3. However, there are descriptions on water*
*balance, wavelet analysis and ARIMA model in sections 2.3.1, 2.3.3 and 2.3.4 respectively. Therefore, we will*
*only revise the trend test section (2.3.2).*

**Action:** *We have written a short over view of the methods in section 2.3 and rewritten the section 2.3.2*
*following the reviewer's suggestions.*

*2.3 Methods*

*In the present study, WBCs have been calculated and trends in WBCs have been identified by MK/MMK test for*
*evaluating the long-term water balance of the highly irrigated western part of Bangladesh. DWT data of WBCs*
*time series has been analyzed for identifying the time period responsible for the trend in the data. WBCs have*
*been forecasted by ARIMA models and the model performance has been evaluated statistically. If the*
*performance of the model is not satisfactory for forecasting the WBCs, the denoising of original time series has*
*been done using discrete wavelet transformation techniques to improve the performance of the model. The*
*descriptions of the methods have been presented in the following sections.*

*2.3.2 Trend Test*

*In the present study, the trends in WBCs have been detected by non-parametric Mann–Kendall (MK) (Mann,*
*1945; Kendal, 1975) test as it shows better performance to identify trends in hydrological variables like rainfall*
*(e.g. Shahid, 2010), temperature (e.g. Kamruzzaman et al., 2016a), $P_{ET}$ (e.g. Kumar et al., 2016), soil moisture*
*(e.g. Tabari and Talaee, 2013), runoff (e.g. Pathak et al., 2016), groundwater level (e.g. Rahman et al., 2016),*
*water quality (e.g. Lutz et al., 2016) in comparison to the parametric test (Nalley et al., 2012). MK test cannot*
*appropriately calculate the test statistic (Z) due to underestimating the variance (Hamed and Rao, 1998) if there*
*is a significant serial correlation at lag-1 in the time series data (Yue et al., 2002). The lag-1 auto-correlation*
*has been checked before analyzing the time series data if there is a significant lag-1 auto-correlation at 5%*
*level, the Modified MK test (Hamed and Rao, 1998) has been applied instead of MK test. The estimated Z*
*statistic of MK/MMK test has been evaluated for the direction of the trend such as positive Z statistic to indicate*
*increasing trend and vice versa. Moreover, it also indicates the level of significance of the obtained trend, for*
*example, if the calculated Z statistic is equal to or greater than the tabulated value of Z statistic +1.96 that*
*indicates a significant positive trend at 95% confidence level or if it is equal to or less than -1.96 that indicates*
*a significant decreasing trend. Moreover, the sequential values of u(t) statistic of MK test derived from the*
*progressive analysis of MK test (Sneyers, 1990), u(t) is similar to the Z statistic (Partal and Küçük, 2006), have*
*been used for investigating the change point detection. The magnitude of the change has been calculated by*
*Sen's slope estimator (Sen, 1968). There are many good explanations (notably Nalley et al., 2012) of these*
*methods mentioned in this section and details regarding these, furthermore, can be referred to Mann (1945);*
*Sen (1968); Kendall (1971); Hamed and Rao (1998); Sneyers (1990); Yue et al. (2002).*

**Comment 8.** Line 154: Which hydrological variables were investigated?

**Reply 8:** *Hydrological variables like rainfall, temperature, $P_{ET}$, runoff, groundwater level and water quality*
*have been investigated by MK test to detect trends in time series data. We have mentioned about these in reply 7*
*(revised section 2.3.2).*

**Action:** *As we mentioned in reply 7, we have rewritten the section 2.3.2.*

**Comment 9.** Line 159: What are these 'Z' values, and what is their importance? This is the first
the time they have been mentioned.

**Reply 9:** *Thank you very much for noticing the Z statistic. We have incorporated text about Z statistic in reply 7*
*(revised section 2.3.2).*

**Action:** *As mentioned earlier, we have rewritten the section 2.3.2 and added necessary text on Z statistic.*

**Comment 10.** Section 2.3.7 is unnecessary here unless it specifically influences the scientific results. Instead put this information in an Appendix or similar (however, I congratulate the authors for putting their computer code alongside the paper – this is not done often enough).

**Reply 10:** *This section will be moved to electronical supplementary material (ESM).*

**Action:** *This section will be moved to electronical supplementary material.*

**Comment 11:** I think Section 3 should simply describe the results, with an additional 'Discussion' section for placing the results in the context of other studies (e.g. Line 281, 288, 321 etc. should be put in a Discussion section). The discussion should include additional analysis discussing the various limitations and weaknesses of the present study as well as suggesting improvements.

**Reply 11:** *We are grateful to you for your valuable comments. We will incorporate a discussion section (3.4) after the results of analysis following your suggestions. We also hope that this section will help readers about the results described in the manuscript and how can we improve the performance of the model.*

**Action:** *We have written a discussion section (3.4) following the suggestions of the reviewers.*

*3.4 Discussion*

*The present study reveals that a decreasing trend in $P_{ET}$ dominates over the study area. However, positive trends in rainfall and temperature dominate in the western part of Bangladesh (e.g. Shahid and Khairulmaini, 2009; Kamruzzaman et al., 2016a). Moreover, a recent study has also found a negative trend in evapotranspiration in four stations located in northwest Bangladesh (Acharjee et al., 2017). Though annual rainfall and temperature of Satkhira station show positive trends (Kamruzzaman et al., 2016a), $P_{ET}$ shows a significant downward trend. Increasing trends in temperature have been found in Yunnan Province of South China, but $P_{ET}$ shows decreasing trend (Fan and Thomas, 2012). McVicar et al. (2012) have also found decreasing trends in $P_{ET}$ in the different parts of the world. Therefore, temperature-based models for the estimation of $P_{ET}$ cannot well explain the causes of changes in $P_{ET}$, though the temperature is the primary driver of changes in $P_{ET}$ (IPCC, 2007). To get a detailed idea about the underlying mechanisms of changes in $P_{ET}$, it is necessary to do a detailed analysis of all climatic variables such as rainfall, temperature, sunshine hours, wind speed, humidity and climate controlling phenomena like El Niño Southern Oscillations (ENSO).*

*The study has also developed WD-ARIMA models for forecasting the WBCs. The performance of the model shows the benefit of denoising of hydrological time series data like $P_{ET}$, $A_{ET}$, surplus and deficit. However, the model performance analysis criterion like NSE indicates that the performance of the model for $P_{ET}$ forecasting is acceptable (NSE $\geq$ 0.65). To have a closer look at the forecasted values and actual values, the deviation between forecast values and actual values increases with increasing time steps. Therefore, WD-ARIMA models are not suitable for long-term forecasting. The present study has developed the WD-ARIMA model by coupling the discrete wavelet denoise time series data and ARIMA model. The soft threshold method has been selected for denoising the time series data and universal threshold (UT) method which has been used for the determination of the threshold value. However, there are some approaches for threshold value determination such as SURE (Stein, 1981), MINMAX (Donoho and Johnstone, 1998) and so on. Moreover, Wang et al. (2014) develop a hybrid approach for denoising the hydro-meteorological time series such as rainfall and streamflow called adaptive wavelet de-noising approach using sample entropy (AWDA-SE). The study has shown that the performance of the developed denoising method is better than conventional de-noising methods for denoising rainfall and streamflow. These approaches may apply to increase the performance of ARIMA models for forecasting hydrological variables like $P_{ET}$. Moreover, there are several mother wavelet families such as*

*Daubechies, Harr, Coiflets, Morlet, Mexican Hat and so on (Sang, 2013). In the present study, only Daubechies-6 from Daubechies wavelet family has been applied as mother wavelet of discrete wavelet transformation. WD-ARIMA models for forecasting the $A_{ET}$, surplus and deficit show very good performance, whereas the classical ARIMA model shows poor performance or unable to forecast the WBCs. Moreover, studies (e.g. Chou, 2011; Kisi, 2008; Partla, 2009; Santos and da Silva, 2014; Rahman and Hasan, 2014; Nury et al., 2016; Adamowski and Chan, 2011; Khalek and Ali, 2016) have also mentioned that the performance of wavelet aided models for forecasting non-stationary hydro-meteorological variables is better than classical ARIMA and ANN models. As the traditional methods such as Wiener filtering, Kalman filtering, Fourier transform are not suitable for non-stationary hydrological time series data (Adamowski and Chan, 2011; Sang, 2013), wavelet denoising can be used to improve the performance of the classical ARIMA models for forecasting hydrological variables.*

**Comment 12**: Line 265: Use 'Potential evapotranspiration' rather than 'Pet' in section headings.

**Reply 12:** *We will incorporate your suggestion. We will replace $P_{ET}$ by Potential Evapotranspiration (3.2.1) and $A_{ET}$ by Actual Evapotranspiration (3.2.2).*

**Action:** *We will replace $P_{ET}$ by Potential Evapotranspiration (3.2.1) and $A_{ET}$ by Actual Evapotranspiration (3.2.2) in the heading.*

**Comment 13.** Line 359: This is just a piece of computer code – what does it do, and what insight does it provide that you cannot gain from manual interpretation of ACF, PACF, AIC, BIC?

**Reply to Comment 13:** *ACF, PACF, AIC, BIC are important parameters for selection of an accurate ARIMA model for forecasting. For manual model sections, we need to find out the best combinations of these parameters with acceptable error. Besides manual model selections, automatic model selection option of the forecast package of R (R-language software) has been used in the present study. This option helps us find out the best model, especially when we could not find a satisfactory model (model with acceptable error) by manual interpretation of ACF, PACF, AIC and BIC.*

**Action:** *We have added the answer here for the reviewer.*

**Comment 14.** Line 362: What is a Q-Q plot?

**Reply 14:** *The quantile-quantile (Q-Q) plot is a probability plot to check the hypothesis of normality for a certain samples. It is graphical method which compares probability distributions based on the quantile values (Filliben, 1975). In our study, we have prepared Q-Q plot to check the normality of residuals.*

**Action:** *We have added the answer here for the reviewer.*

**Comment 15.** Line 386-416: To my mind this passage is the strongest part of the paper -the discussion should emphasise this result and its relevance to water resources management more generally.

**Reply to Comment 15:** *Thank you very much again for your observations and comments. We will add a discussion section as we have mentioned and added in reply 11.*

**Action:** *As mentioned earlier, we have added a discussion section (3.4). Please go to the reply 11.*

**Comment 16:** As I mentioned earlier, I would strongly suggest creating an additional Discussion section in which to discuss the results in the context of other studies, highlight limitations and propose future research directions.

**Reply to Comment 16:** *We have mentioned the matter earlier. We are grateful to you for your comments that help us improve the quality of our present research work. Thank you very much again.*

**Action:** *As mentioned earlier, we have added a discussion section (3.4). Please go to the reply 11.*

[revised manuscript text omitted]

Sang, Y.F. *A review on the applications of wavelet transform in hydrology time series analysis. Atmospheric Research, 122, 8-15, 10.1016/j.atmosres.2012.11.003, 2013.*

Santos, C.A.G & da Silva, G. B. L. *Daily streamflow forecasting using a wavelet transform and artificial neural network hybrid models, Hydrological Sciences Journal, 59:2, 312-324, DOI: 10.1080/02626667.2013.800944, 2014.*

Sen, P. K. *Estimates of the regression coefficient based on Kendall's tau, Journal of the American Statistical Association, 63(324), 1379–1389, 1968.*

Shahid, S. and Khairulmaini, O. S. *Spatial and temporal variability of rainfall in Bangladesh, Asia-pacific Journal of Atmospheric Sciences, 45(3), 375–389, 2009.*

Shahid, S. *Recent trends in the climate of Bangladesh. Climatic Research, 42(3), 185–193, 2010.*

Sneyers, R. *On the Statistical Analysis of Series of Observations, Secretariat of the World Meteorological Organization, (192 pp), 1990.*

SRDI (Soil Resources Development Institute). *Soil map of Bangladesh, Soil Resources Development Institute, 1998.*

Stein C. M. *Estimation of the Mean of a Multivariate Normal-Distribution. Annals of Statistics, Vol. 9, No. 6, (November 1981), pp. 1317-1322, ISSN 0090-5364, 1981.*

Syed, A. and Al Amin. *Geospatial Modeling for Investigating Spatial Pattern and Change Trend of Temperature and Rainfall. Climate, 4, 21, doi:10.3390/cli4020021, 2016.*

Tabari, H. and Talaee, P.H. *2013. Moisture index for Iran: Spatial and temporal analyses, Global and Planetary Change, 100: 11-19.*

Thornthwaite, C. W. *An approach towards a rational classification of climate, Geographical Review, 38, 55–94, 1948.*

Thornthwaite, C. W., and Mather, J. R. *The Water Balance, Publications in Climatology VIII(1): 1-104, Drexel Institute of Climatology, Centerton, New Jersey . 1955.*

Thornthwaite, C. W. and Mather, J. R. *Instructions and tables for computing potential evapotranspiration and the water balance. Publications in Climatology, 10(3), 183–311, 1957. Laboratory of Climatology, Drexel Institute of Technology, Centerton, New Jersey, USA.*

Tiwari, M.K., Chatterjee, C., 2010. *Development of an accurate and reliable hourly flood forecasting model using wavelet–bootstrap–ANN (WBANN) hybrid approach. Journal of Hydrology 1 (394), 458–470.*

Valipour, M. *Ability of Box-Jenkins Models to Estimate of Reference Potential Evapotranspiration (A Case Study: Mehrabad Synoptic Station, Tehran, Iran), IOSR Journal of Agriculture and Veterinary Science, 1(5), 01-11. 2012.*

Viaroli, S., Mastrorillo, L., Lotti, F., Paolucci, V., Mazza. R. *The groundwater budget: a tool for preliminary estimation of the hydraulic connection between neighboring aquifers. Journal of hydrology, doi.org/10.1016/j.jhydrol.2017.10.066, 2017.*

Wang, D., Singh, V. P., Shang, X., Ding, H., Wu, J., Wang, L., Zou, X. Chen, Y., Chen, X., Wang, S. and Wang, Z. *Sample entropy based adaptive wavelet de-noising approach for meteorological and hydrologic time series, Journal of Geophysical Research and Atmosphere, 119, 8726–8740, doi:10.1002/ 2014JD021869, 2014.*

Wolock, D. M. and McCabe, G. J. *Effects of potential climatic change on annual runoff in the conterminous United States, Journal of the American Water Resources Association, 35, 1341–1350, 1999.*

Xu, C. -Y. and Halldin, S. *The effect of climate change in river flow and snow cover in the NOPEX area simulated by a simple water balance model, Proc. of Nordic Hydrological Conference, Alkureyri, Iceland, 1, 436–445, 1996.*

Xu, C. -Y. and Singh, V. P. *Cross comparison of empirical equations for calculating potential evapotranspiration with data from Switzerland, Water Resources Management, 16, 197–219, 2002.*

[revised manuscript text omitted]

---

## Author Response (AR2)

Anonyms Referee # 2

We are very much grateful to the anonym's referee (#2) for reviewing our manuscript second time. We have responded to referee (#2) comments below**:**

**Comment 1:**

I'm grateful to the authors for taking my previous comments into account and I believe the revised manuscript is a substantial improvement which sufficiently addresses my comments on the scientific content. It is obvious the authors' have spent time working on the quality of the English and indeed this aspect of the paper is much improved. However, there are still some passages which are confusing and distract from the underlying science. Therefore may I suggest that before the final submission they have the manuscript proofread again, preferably by a third party with full professional proficiency in written English.

**Reply1:**

We have followed your suggestions. We have done the manuscript proofreading by a professional proofreading company to correct the English language. We hope that there is no problem related to the quality of the English in the manuscript.   Thank you very much again for your constructive comments and suggestions.

**Modeling the changes in water balance components of highly irrigated western part of Bangladesh**

A.T.M. Sakiur Rahman[1*], Md. Shakil Ahmed[2], Hasnat Mohammad Adnan[3], Mohammad Kamruzzaman[4], Md. Abdul Khalek[2], Quamrul Hasan Mazumder[3], and Chowdhury Sarwar Jahan[3]

**Author details:**

[1]Hydrology Lab, Dept. of Earth and Environmental Sciences, Graduate School of Science and Technology, Kumamoto University, 2-40-1 Kurokami, Kumamoto, Japan
[2]Department of Statistics, University of Rajshahi, Rajshahi 6205, Bangladesh
[3]Department of Geology and Mining, University of Rajshahi, Rajshahi 6205, Bangladesh
[4]Institute of Bangladesh Studies, University of Rajshahi, Rajshahi 6205, Bangladesh

*Corresponding author e-mail: shakigeo@gmail.com

**Abstract:** The objectives of the present study were to explore the changes in the water balance components (WBCs) by co-utilizing the discrete wavelet transform (DWT) and different forms of the Mann–Kendal (MK) test and develop a wavelet denoise autoregressive integrated moving average (WD-ARIMA) model for forecasting the WBCs. The results revealed that most of the potential evapotranspiration ($P_{ET}$) trends (approximately 73%) had a decreasing tendency from 1981–82 to 2012–13 in the western part of Bangladesh. However, most of the trends (approximately 82%) were not statistically significant at a 5% significance level. The actual evapotranspiration ($A_{ET}$), annual deficit, and annual surplus also exhibited a similar tendency. The rainfall and temperature exhibited increasing trends. However, the WBCs exhibited an inverse trend, which suggested that the $P_{ET}$ changes associated with temperature changes could not explain the change in the WBCs. Moreover, the 8-year (D3) and 16-year (D4) periodic components were generally responsible for the trends found in the original WBC data for the study area. The actual data was affected by noise, which resulted in the ARIMA model exhibiting an unsatisfactory performance. Therefore, wavelet denoising of the WBC time series was conducted to improve the performance of the ARIMA model. The quality of the denoising time series data was ensured using relevant statistical analysis. The performance of the WD-ARIMA model was assessed using the Nash–Sutcliffe efficiency (NSE) coefficient and coefficient of determination ($R^2$). The WD-ARIMA model exhibited very good performance, which clearly demonstrated the advantages of denoising the time series data for forecasting the WBCs. The validation results of the model revealed that the forecasted values were very close to actual values, with an acceptable mean percentage error. The residuals also followed a normal distribution. The performance and validation results indicated that models can be used for the short-term forecasting of WBCs. Further studies on different combinations of wavelet analysis are required to develop a superior model for the hydrological forecasting in context of climate change. The findings of this study can be used to improve water resource management in the highly irrigated western part of Bangladesh.

**Keywords:** Discrete wavelet transformation, Wavelet denoising, Water balance, ARIMA model

**1. Introduction**

The monthly water balance model of Thronthwaite (1948) was modified by Thornthwaite and Mather (1957). This model has been further modified for application in different areas of the world. Improvements are still being made to the water balance model (Xu and Singh, 1998) because the 
[revised manuscript text omitted]

---

## Author Response (AR3)

**Reply to Editor Comments:**

**Editor Decision: Publish subject to technical corrections** (09 Jul 2018) by Ana Mijic

**Reply:** Thank you very much for accepting our manuscript for publishing in the journal of Hydrology and Earth System Sciences.

Comments to the Author:

Line 71. At the beginning of the Introduction please include introduction sentence on the importance of the water balance modelling to introduce the topic (or rearrange some of the text below).

**Reply 1:** We have rearranged some sentences at the beginning of the Introduction following your instructions.

Line 78. Please define the WBC abbreviation in the main text.

**Reply 2:** We have also incorporated this correction.